# AlignedNorm: Prompting Vision–Language Models via Coupled Prompt Field

Qi Ma [1 2]  Chen-Yang Wang [1 2]  Dehong Gao [3]  Deng-Ping Fan [1 2 4]

## Abstract

Prompt learning for vision-language models (VLMs) primarily follows end-to-end or decoupled routes to balance base and new task performance, but suffers a fundamental bottleneck: sample-wise optimization within task-specific feature spaces traps models in local optima, hindering global optimality. To address this, we identify a key insight that VLMs can be prompted within a **Coupled Prompt Field** - a shared space where base and new tasks are *mutually constrained* - and present **AlignedNorm**, which enforces the field coupling. By dynamically aligning the norms of prompts to VLMs' native scale, our method enables joint optimization of both tasks. Without complex designs, our method matches leading decoupled approaches on 15 datasets across 4 experimental settings, offering both a new perspective and a practical solution to the local-optimum dilemma in prompt learning. Code is available at https://github.com/QByteM/AlignedNorm.

## 1. Introduction

Prompt learning can efficiently adapt vision-language models (VLMs), such as Contrastive Language-Image Pre-training (CLIP) (Radford et al., 2021), to a broad range of downstream tasks. Recent progress has been driven by methods that seek a better trade-off between adaptability and generalization. Existing approaches can be broadly grouped into two paradigms: end-to-end prompt learning (Zhou et al., 2022a; Khattak et al., 2023b;a) and decoupled prompt learning (Zhang et al., 2024; Li et al., 2025a; Guo & Gu, 2025).

Despite their success, both paradigms still struggle to balance base-task adaptation with new-task generalization. We attribute this in part to *isolated* prompt optimization, in which the learned prompt-induced change is tied to task-

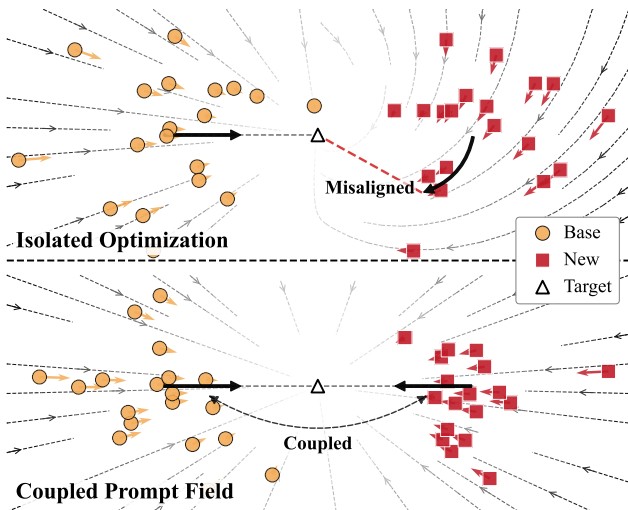

*Figure 1.* **Our coupled prompt field.** We couple Base and New tasks by learning the prompt-induced change as a shared field. Despite the gap between Base and New, this field provides a task-agnostic correction that mutually constrains the two tasks, bringing them toward a common objective under a unified goal.

specific feature spaces. This motivates us to view prompt learning as *Field Construction*: learning a task-shared transformation over the feature space that can couple base and new tasks under a unified inference rule.

We argue that the prompt-induced field should be coupled across base and new tasks so that the two are mutually constrained rather than optimized in isolation (see Fig. 1). However, many recent methods (Zhang et al., 2024; Guo & Gu, 2025; 2026) still rely on task-specific optimization or inference mechanisms, which can weaken task-agnostic coupling and limit reliable cross-task generalization.

To this end, we propose a framework that instantiates prompt learning via a **Coupled Prompt Field** (CPF). This raises a natural question: *what is essential for constructing such a coupled field?* We observe that norm drift disrupts the uniformity tolerance balance (Wang & Isola, 2020; Wang & Liu, 2021) maintained by VLMs, which undermines task-agnostic coupling. This makes dynamic norm alignment necessary for a coupled prompt field. Building on our coupled-field analysis (see §4.1), we identify the embedding norm as the answer and introduce a simple norm-based regularizer, **AlignedNorm**, to constrain the tuning dynamics.

[1]VCIP & CS, Nankai University [2]NKIARI, Shenzhen Futian [3]Northwestern Polytechnical University [4]SLAI. Correspondence to: Deng-Ping Fan <fdp@nankai.edu.cn>.

*Proceedings of the 43rd International Conference on Machine Learning*, Seoul, South Korea. PMLR 306, 2026. Copyright 2026 by the author(s).

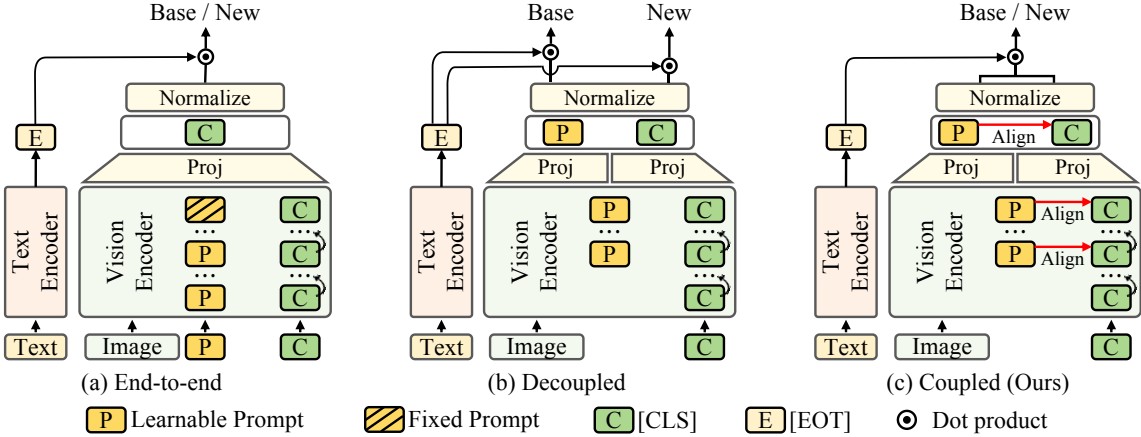

*Figure 2.* **Comparison of prompt-learning paradigms for VLMs.** (a) *End-to-end*: prompts are optimized on base data and directly reused for both base and new tasks, often inducing a base–new trade-off. (b) *Decoupled*: adaptation and inference are separated (*e.g.*, different branches for base vs. new), typically requiring the task identity at test time. (c) *Coupled (Ours)*: we construct a *Coupled Prompt Field* by coupling the prompt pathway via norm alignment in each encoder layer and at the projected features.

Post-projection norm alignment may be insufficient when prompt-token interactions are already weakened in intermediate transformer layers. In our experiments, randomly initialized prompts can exhibit weak attention coupling with global tokens, a phenomenon we refer to as *entanglement collapse*, resulting in failed global information exchange. **AlignedNorm** therefore aligns prompt-token norms with class-token norms at each prompted layer so that the coupled field remains stable throughout the encoder.

Without external knowledge distillation or decoupled inference, AlignedNorm delivers highly competitive performance and demonstrates robust generalization capabilities across the evaluated settings. Uniformity and tolerance metrics further show that the geometry gap observed in MMRL++ (Guo & Gu, 2026) can be mitigated by Aligned-Norm, providing a complementary explanation for the effectiveness of our method.

In summary, we formulate VLM prompt learning through a CPF, which models the prompt-induced transformation as a task-shared field where base and new tasks are mutually constrained. This formulation preserves pretrained knowledge as a stable task-shared anchor while allowing expressive task-specific adaptation under a unified inference rule. We further identify embedding norm as a key factor and introduce AlignedNorm to stabilize this coupling. Our method combines an intuitive post-projection norm alignment to stabilize the final prompt-induced field with layer-wise norm alignment to alleviate potential entanglement collapse in intermediate prompt-token interactions. Across 15 datasets and 4 evaluation settings, AlignedNorm achieves highly competitive performance. We hope our CPF formulation and norm-alignment insights inspire new directions in efficient, generalizable multi-modal prompt learning.

## 2. Related Work

**End-to-end prompt learning.** Early works (Zhou et al., 2022a;b; Khattak et al., 2023a) adapt models through context optimization. These methods suffer from critical limitations, most notably the Base-New Tradeoff, *i.e.*, the better the adapted model performs on base tasks, the worse it generalizes to new tasks with unseen classes.

Subsequently, mainstream methods have focused on mitigating this tradeoff, mainly falling into three categories. The most widely used method is ***(i) self-regularization***: this approach mitigates overfitting by leveraging the inherent knowledge of the original CLIP model (Yao et al., 2023; Khattak et al., 2023b; Xie et al., 2025), and regularization can also be effectively achieved through meta-learning strategies (Park et al., 2024). As self-regularization signals saturate, ***(ii) external reference*** supplements them: one sub-type enhances generalization by mimicking powerful teacher models (Li et al., 2024c), while the other (Ding et al., 2025; Khattak et al., 2025; Li et al., 2025b;c) uses large models to construct discrete anchors for optimization guidance; these methods yield significant performance gains but incur large additional overhead. The third category relaxes the model's final outputs through ***(iii) uncertainty modeling*** to avoid overfitting from overly absolute single solutions (Lu et al., 2022; Cheng & Han, 2025).

These methods seek to attain predefined ideal results through an end-to-end paradigm, yet the inherent gap between base and new tasks makes predefined ideal results undefinable.

**Decoupled prompt learning.** Recently, several methods have recognized the interference of knowledge learned from base tasks on new tasks and have proposed modeling the knowledge of these two types of tasks separately-

typically by learning task-specific knowledge in an isolated feature space to preserve the original generalization ability of the model. Specifically, DePT (Zhang et al., 2024) achieves knowledge isolation through a subnetwork, DPC (Li et al., 2025a) accomplishes it via different prompts, and MMRL (Guo & Gu, 2025) as well as MMRL++ (Guo & Gu, 2026) realize this goal by establishing distinct representation spaces. These methods preserve pretrained knowledge and mitigate catastrophic forgetting, but they have a notable limitation: their operation requires prior identification of whether the current task is a base task or a new task, which is often impractical in open-world settings.

Existing paradigms focus on optimizing final outputs for dual-task satisfaction. In contrast, our work analyzes the prompt process from a dynamic perspective, which requires no predefined ideal results and emphasizes beneficial changes, offering new insights for method extension.

## 3. Preliminaries

### 3.1. End-to-end Prompt Learning

Prompt Learning (Zhou et al., 2022a; Khattak et al., 2023a; Guo & Gu, 2025) adapts CLIP by introducing learnable tokens into the visual encoder, the text encoder, or both. Formally, given an input image $x$, the CLIP visual encoder outputs a class embedding from the $L$-th layer $c_L(x) \in \mathbb{R}^D$, which is mapped to the shared image-text space by the visual projection head for class-token $P_v^c : \mathbb{R}^D \to \mathbb{R}^d$: $z_c(x) = P_v^c(c_L(x)) \in \mathbb{R}^d$. For $C$ classes, the corresponding $\ell_2$-normalized text class features are denoted as $\{w_k\}_{k=1}^C$ with $w_k \in \mathbb{R}^d$. The image embedding is further $\ell_2$-normalized as $f_c(x) = \frac{z_c(x)}{\|z_c(x)\|}$, and the temperature-scaled classifier $\pi(\cdot)$ with $\tau > 0$ is defined by $\pi(f) = \mathrm{Softmax}\left(\frac{1}{\tau}\left[w_k^\top f\right]_{k=1}^C\right)$, where $\pi_k(f)$ denotes the predicted probability of the $k$-th class. Given a training sample $(x, y)$, optimization is performed using the cross-entropy loss $\mathcal{L}_{ce}(\pi(f_c(x)), y)$.

### 3.2. Decoupled Prompt Learning

Latest decoupled prompt learning (Guo & Gu, 2025; 2026) extends prompt learning by injecting learnable prompts into deeper layers of the CLIP encoder and using the pooled prompt tokens as an additional classification branch. Let $\mathcal{V} = \{\mathcal{V}_i\}_{i=1}^L$ denote a $L$-layer ViT visual encoder with hidden width $D$. At layer $i$, the class token is $c_i \in \mathbb{R}^D$, the patch tokens are $E_i \in \mathbb{R}^{M \times D}$, and the prompt tokens are $P_i \in \mathbb{R}^{n_p \times D}$. Prompts are activated from a designated layer $J$ onward:

$$[c_i, E_i] = \mathcal{V}_i([c_{i-1}, E_{i-1}]), \quad i = 1, \dots, J-1,$$
$$[c_j, P_j, E_j] = \mathcal{V}_j([c_{j-1}, \widetilde{P}_j, E_{j-1}]), \quad j = J, \dots, L, \quad (1)$$

where $\widetilde{P}_j \in \mathbb{R}^{n_p \times D}$ denotes the prompt input of layer $j$. After the final layer, prompt tokens are aggregated by

mean pooling, $\bar{P}_L(x) = \mathrm{Mean}(P_L(x))$, and projected to the shared space by $P_v^p : \mathbb{R}^D \to \mathbb{R}^d$ as $z_p(x) = P_v^p(\bar{P}_L(x))$, followed by $\ell_2$ normalization $f_p(x) = \frac{z_p(x)}{\|z_p(x)\|}$. Together with the class-token embedding $f_c(x)$ defined previously, both branches are classified by the same classifier $\pi(\cdot)$.

**Visual anchoring.** Visual anchoring is implemented via self-regularization (Khattak et al., 2023b; Roy & Etemad, 2024) and applied only to $f_c$, thereby decoupling the two branches: $f_c$ preserves CLIP semantics while $f_p$ focuses on task-specific adaptation.

**Task objective.** Given a training sample $(x, y)$, the method is optimized by $\mathcal{L}_{\mathrm{task}}$ (Guo & Gu, 2025; 2026).

**Decoupled inference.** For base classes, the two branches (Guo & Gu, 2025; 2026) are combined as:

$$f_b(x) = (1 - \alpha) f_c(x) + \alpha f_p(x), \quad (2)$$

and predictions are obtained by $\hat{p}_k(x) = \pi_k(f_b(x))$; for new classes, $\hat{p}_k(x) = \pi_k(f_c(x))$.

## 4. Method

### 4.1. Proposed Coupled Prompt Field

**Motivation.** Existing prompt learning methods fail to predefine the ideal final results, mainly due to the inherent base-new gap. Inspired by classic perspectives (He et al., 2016; Chen et al., 2018; Lipman et al., 2023; Ilievski et al., 2025), we consider that the prompt-induced change is more transferable across the base-new gap. Based on the advanced decoupled deep prompt learning framework (Guo & Gu, 2026), we are able to explicitly model this change as a field over the feature space, which leads to a single task-agnostic inference rule.

**Field construction.** Decoupled inference in Eqn. 2 forms a mixed embedding for base classes. Rewriting it in an anchored form makes the structure explicit:

$$f_b(x) = f_c(x) + \alpha(f_p(x) - f_c(x)). \quad (3)$$

Due to the visual anchoring, $f_c(x)$ can be viewed as a stable anchor over the feature space. We therefore define the prompt field as:

$$\mathbf{u}_f(x) \triangleq \alpha(f_p(x) - f_c(x)) \in \mathbb{R}^d. \quad (4)$$

**Coupled prompt field.** A key practical limitation of decoupled learning is that the task identity (base vs. new) is typically unavailable in open-world evaluation. The field view suggests a single inference strategy for all tasks: $\hat{p}_k(x) = \pi_k(f_c(x) + \mathbf{u}_f(x))$. This turns decoupled inference into a task-agnostic coupling requirement: the same field $\mathbf{u}_f(\cdot)$ must generalize across the base-new gap.

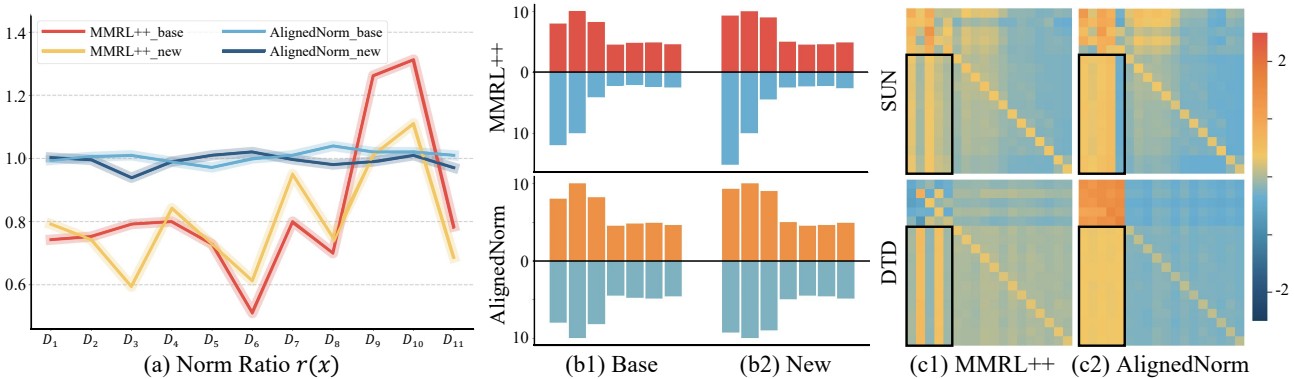

*Figure 3.* **Empirical observations.** (a) Visualization of $r(x)$ across 11 datasets for MMRL++ (Base and New) and AlignedNorm (Base and New). (b) Statistics of token norm in 7 layers on ImageNet for MMRL++ (class and prompt) and AlignedNorm (class and prompt). (c) Visualization of attention logits on two datasets (the first five tokens are prompt tokens). MMRL++ shows isolated prompt tokens with weak attention, while AlignedNorm restores prompt-patch coupling for better global information exchange.

**Field distortion.** Under coupled inference, define the norm ratio $r(x) \triangleq \frac{\|z_p(x)\|}{\|z_c(x)\|}$. Consequently, the coupled field depends on the sample-wise ratio $r(x)$:

$$\mathbf{u}_f(x) = \alpha \left( \frac{z_p(x)}{\|z_p(x)\|} - \frac{z_c(x)}{\|z_c(x)\|} \right)$$
$$= \frac{\alpha}{\|z_c(x)\|} \left( r(x)^{-1} z_p(x) - z_c(x) \right). \quad (5)$$

As shown in Fig. 3(a), the norm ratio $r(x)$ in MMRL++ shows an obvious base-new gap: its values on $\mathcal{D}_{\text{base}}$ and $\mathcal{D}_{\text{new}}$ are clearly separated. Such a shift in $r(x)$ implies that the prompt-induced correction is inconsistently scaled under the same coupled rule, leading to unintended distortions of the coupled prompt field across the base–new gap:

$$\mathbb{E}_{x \sim \mathcal{D}_{\text{base}}}[r(x)] \neq \mathbb{E}_{x \sim \mathcal{D}_{\text{new}}}[r(x)], \quad (6)$$

and thus violating the task-agnostic and global consistency.

**Non-uniform field updates.** Consider any loss term $\mathcal{L}$ that depends on the prompt branch only through the normalized embedding $f_p(x)$. Denote $g(x) \triangleq \partial \mathcal{L}/\partial f(x) \in \mathbb{R}^d$. Backpropagation gives:

$$\frac{\partial \mathcal{L}}{\partial z(x)} = \frac{1}{\|z(x)\|} \left( I - f(x)f(x)^\top \right) g(x). \quad (7)$$

Hence, the gradient magnitude is inversely scaled by $\|z(x)\|$. Assume the stochastic gradient admits a decomposition $g(x) = \bar{g}(x) + \xi(x)$ with $\mathbb{E}[\xi(x)] = 0$. Then the update noise on the prompt branch is amplified as:

$$\mathbb{E}\left[ \left\| \nabla_z \mathcal{L}(x) - \mathbb{E}[\nabla_z \mathcal{L}(x)] \right\|^2 \right] \leq \mathbb{E}\left[ \frac{\|\xi(x)\|^2}{\|z(x)\|^2} \right]. \quad (8)$$

Therefore, when $\|z_p(x)\|$ is smaller than $\|z_c(x)\|$, the $1/\|z_p(x)\|$ factor in Eqn. 8 amplifies the effective update

scale on the prompt branch, increasing its sensitivity to stochastic gradient noise (see Appendix B.1).

**Field stability to perturbations.** Consider a stochastic perturbation on the prompt branch, $z_p = z_p^\star + \varepsilon$, $\mathbb{E}[\varepsilon \mid x] = 0$, where $\varepsilon$ models estimation noise induced by finite data and stochastic optimization. Using the first-order around $z_p^\star$, $f_p(x) - f_p^\star(x) \approx \frac{1}{\|z_p^\star(x)\|} \left( I - f_p^\star(x) f_p^{\star\top}(x) \right) \varepsilon$, we have the following bound:

$$\mathbb{E}\left[ \|\mathbf{u}_f(x) - \mathbf{u}_f^\star(x)\|_2^2 \mid x \right] \approx \alpha^2 \, \mathbb{E}\left[ \|f_p(x) - f_p^\star(x)\|_2^2 \mid x \right]$$
$$\leq \alpha^2 \frac{\mathbb{E}[\|\varepsilon\|_2^2 \mid x]}{\|z_p^\star(x)\|^2}. \quad (9)$$

Anchoring keeps $f_c$ relatively stable, so controlling $\|z_p(x)\|$ is key to improving the noise robustness of a coupled prompt field. (see Appendix B.2)

### 4.2. Proposed AlignedNorm

Eqn. 5 and Eqn. 6 show that base-new $r(x)$ inconsistency distorts the coupled prompt field. Eqn. 8 and Eqn. 9 further reveal that the prompt-branch norm affects the uniformity and stability of the coupled prompt field. Together, these results indicate that stabilizing the projected prompt norm is key to robust task-agnostic coupling.

**Norm alignment after projection.** We therefore directly regularize the final projected prompt feature by enforcing its norm to align with the class token norm via an $\ell_1$ penalty:

$$\mathcal{L}_{\text{proj}}(x) = \left| \, \|z_p(x)\| - \text{sg}(\|z_c(x)\|) \, \right|, \quad (10)$$

where $\text{sg}(\cdot)$ stops gradients. However, alignment after projection yields limited gains. This motivates a deeper analysis of the underlying norm dynamics within the visual encoder.

**Entanglement collapse.** Recall that attention logits are driven by dot products, which can be written as $q^\top k =$

**Algorithm 1** AlignedNorm Loss

---

1: **Input:** an image $x$, class tokens $\{c_l(x)\}_{l=J}^{L}$, prompt tokens $\{P_l(x)\}_{l=J}^{L}$, projected outputs $z_c(x), z_p(x)$, weights $\beta, \gamma$
2: **Output:** alignment loss $\mathcal{L}_{\text{align}}$
3: *// 1. Norm alignment after projection*
4: $\quad n_{z_c} \leftarrow \|z_c(x)\|, \quad n_{z_p} \leftarrow \|z_p(x)\|$
5: $\quad \mathcal{L}_{\text{proj}} \leftarrow \left| n_{z_p} - \text{sg}(n_{z_c}) \right|$
6: Initialize $\mathcal{L}_{\text{token}} \leftarrow 0$
7: *// 2. Norm alignment in each layer*
8: **for** $l = J$ **to** $L$ **do**
9: $\quad n_c \leftarrow \|c_l(x)\|, \quad n_p \leftarrow \|\text{Mean}(P_l(x))\|$
10: $\quad \mathcal{L}_{\text{token}} \leftarrow \mathcal{L}_{\text{token}} + \left| n_p - \text{sg}(n_c) \right|$
11: **end for**
12: $\mathcal{L}_{\text{align}} \leftarrow \beta \, \mathcal{L}_{\text{proj}} + \gamma \, \mathcal{L}_{\text{token}}$
13: **return** $\mathcal{L}_{\text{align}}$

---

$\|q\| \, \|k\| \cos\theta$. In high dimensions, randomly initialized prompt token directions are nearly orthogonal with high probability, so $\cos\theta$ is close to 0 at early training stages (see Appendix B.3), leading to weak and easily imbalanced prompt–patch logits. We find that prompt tokens exhibit larger norms than the class token during training (see Fig. 3 (b)), which enlarges the logit gap under misalignment and quickly saturates softmax gating into a one-sided regime that is difficult to recover from. Consequently, many global tokens assign near-zero attention to prompt tokens (see Fig. 3 (c)), causing an entanglement collapse where prompts fail to exchange semantic information with the global representation. (see Appendix B.4)

**Norm alignment in each layer.** Alignment in Eqn. 10 stabilizes the output norm $\|z_p(x)\|$, but does not directly control the prompt-token scale inside the transformer. We therefore align prompt-token scales to the class-token in each prompted layer. For $l = J, \ldots, L$, we define the layer-wise pooled prompt token $\bar{P}_l(x) \triangleq \text{Mean}(P_l(x))$ and impose

$$\mathcal{L}_{\text{token}} = \sum_{l=J}^{L} \left| \|\bar{P}_l(x)\| - \text{sg}\big(\|c_l(x)\|\big) \right|. \tag{11}$$

**Overall objective.** We combine norm alignments into a single loss (see Algorithm 1):

$$\mathcal{L}_{\text{align}} = \beta \, \mathcal{L}_{\text{proj}} + \gamma \, \mathcal{L}_{\text{token}}, \tag{12}$$

where $\beta$ and $\gamma$ control their relative strengths; the overall training objective is $\mathcal{L} = \mathcal{L}_{\text{task}} + \mathcal{L}_{\text{align}}$.

# 5. Experiment

## 5.1. Evaluation Settings

Following prior works (Guo & Gu, 2025; 2026), we evaluate AlignedNorm's capabilities on four benchmarks:

**Datasets.** We evaluate AlignedNorm on 11 datasets to probe its generalizability across diverse task scenarios, spanning generic object recognition (ImageNet (Deng et al., 2009), Caltech101 (Fei-Fei et al., 2004)), fine-grained classification (OxfordPets (Parkhi et al., 2012), StanfordCars (Krause et al., 2013), Flowers102 (Nilsback & Zisserman, 2008), Food101 (Bossard et al., 2014), and FGVCAircraft (Maji et al., 2013)), action recognition (UCF101 (Soomro et al., 2012)), texture classification (DTD (Cimpoi et al., 2014)), scene recognition (SUN397 (Xiao et al., 2010)), and satellite image classification (EuroSAT (Helber et al., 2019)). We further assess AlignedNorm's robustness to domain shift on ImageNet-A (Hendrycks et al., 2021b), ImageNet-R (Hendrycks et al., 2021a), ImageNet-Sketch (Wang et al., 2019), and ImageNetV2 (Recht et al., 2019), which together provide a broad evaluation of generalization.

**Base-to-new generalization.** To assess the open-set generalizability of the model within a dataset, each dataset is evenly split into two class-disjoint subsets: base and new. We train exclusively on base and test on both splits. This task demands a balance between learning capacity and class-level generalization, quantified by the harmonic mean (HM) of accuracy on the base and new classes.

**Cross-dataset transfer.** In this setting, the source model is trained on all 1,000 categories of ImageNet (Deng et al., 2009) and is directly evaluated on each of the remaining 10 datasets without any fine-tuning.

**Few-shot learning.** This benchmark aims to probe the model's learning capacity under data scarcity, *i.e.*, training the models with K-shots (K = {1, 2, 4, 8, 16}) of each dataset and testing on full test set. The model is tasked with acquiring task-specific knowledge while preserving CLIP's pretrained representation.

**Domain generalization.** The robustness of the model is evaluated on specially designed out-of-distribution datasets. The source model is trained on all categories of ImageNet (Deng et al., 2009) and generalized on four datasets, each characterized by a distinct domain shift.

**Implementation details.** AlignedNorm is built upon the MMRL++ model, and the results reported are weighted averages over 3 random seeds. The experimental settings are consistent with the default configuration of MMRL++. Additional implementation details can be found in Appendix C. All experiments are conducted on an RTX 3090 GPU.

## 5.2. Base-to-new Generalization Experiment

As shown in Table 1, we consider several representative methods from both the end-to-end and decoupled paradigms. We compare against MaPLe (Khattak et al., 2023a), Prompt-SRC (Khattak et al., 2023b), and HicroPL (Zheng et al., 2025) for the end-to-end setting. For the decoupled strat-

*Table 1.* **Base to new generalization experiment.** The accuracy (%) on base and new classes for each dataset, and HM denotes the harmonic mean. Δ indicates the improvement achieved by AlignedNorm compared with MMRL++ without decoupling strategy.

| Method | Average | | | ImageNet | | | Caltech101 | | | OxfordPets | | |
|---|---|---|---|---|---|---|---|---|---|---|---|---|
| | Base | New | HM | Base | New | HM | Base | New | HM | Base | New | HM |
| CLIP (Radford et al., 2021) | 69.34 | 74.22 | 71.70 | 72.43 | 68.14 | 70.22 | 96.84 | 94.00 | 95.40 | 91.17 | 97.26 | 94.12 |
| *Methods based on the* **end-to-end** *paradigm:* | | | | | | | | | | | | |
| MaPLe (Khattak et al., 2023a) | 81.91 | 75.09 | 78.35 | 76.60 | 70.77 | 73.57 | 97.73 | 95.30 | 96.50 | 95.70 | 98.07 | 96.87 |
| PromptSRC (Khattak et al., 2023b) | 84.23 | 75.78 | 79.78 | 77.60 | 70.37 | 73.81 | 98.07 | 94.00 | 95.99 | 95.23 | 97.17 | 96.19 |
| HicroPL (Zheng et al., 2025) | 85.16 | 76.50 | 80.60 | 78.34 | 71.68 | 74.86 | 98.36 | 95.45 | 96.88 | 95.62 | 97.69 | 96.64 |
| *Methods based on the* **decoupled** *paradigm:* | | | | | | | | | | | | |
| MMRL (Guo & Gu, 2025) | 85.71 | 76.28 | 80.72 | 77.70 | 71.20 | 74.31 | 98.83 | 94.33 | 96.53 | 95.97 | 97.50 | 96.73 |
| w/o decoupling strategy | 85.71 | 72.60 | 78.61 | 77.70 | 69.13 | 73.16 | 98.83 | 94.83 | 96.79 | 95.97 | 94.70 | 95.33 |
| MMRL++ (Guo & Gu, 2026) | 85.43 | 77.79 | 81.43 | 77.60 | 71.40 | 74.37 | 98.90 | 94.40 | 96.60 | 95.43 | 96.97 | 96.19 |
| w/o decoupling strategy | 85.43 | 76.48 | 80.71 | 77.60 | 71.30 | 74.32 | 98.90 | 94.57 | 96.69 | 95.43 | 96.87 | 96.14 |
| *Methods based on the* **coupled prompt field**: | | | | | | | | | | | | |
| **AlignedNorm** | 85.46 | 77.79 | 81.45 | 77.60 | 71.47 | 74.41 | 98.90 | 94.77 | 96.79 | 95.63 | 97.43 | 96.52 |
| Δ | **+0.03** | **+1.31** | **+0.74** | 0.00 | **+0.17** | **+0.09** | 0.00 | **+0.20** | **+0.10** | **+0.20** | **+0.56** | **+0.38** |

| Method | StanfordCars | | | Flowers102 | | | Food101 | | | FGVCAircraft | | |
|---|---|---|---|---|---|---|---|---|---|---|---|---|
| | Base | New | HM | Base | New | HM | Base | New | HM | Base | New | HM |
| CLIP (Radford et al., 2021) | 63.37 | 74.89 | 68.65 | 72.08 | 77.80 | 74.83 | 90.10 | 91.22 | 90.66 | 27.19 | 36.29 | 31.09 |
| *Methods based on the* **end-to-end** *paradigm:* | | | | | | | | | | | | |
| MaPLe (Khattak et al., 2023a) | 72.30 | 73.80 | 73.04 | 96.03 | 73.33 | 83.16 | 90.70 | 92.03 | 91.36 | 36.07 | 34.47 | 35.25 |
| PromptSRC (Khattak et al., 2023b) | 78.20 | 75.47 | 76.81 | 98.07 | 77.37 | 86.50 | 90.63 | 91.50 | 91.06 | 43.33 | 36.27 | 39.49 |
| HicroPL (Zheng et al., 2025) | 81.13 | 75.04 | 77.97 | 98.10 | 74.75 | 84.85 | 90.74 | 91.72 | 91.23 | 46.06 | 37.61 | 41.41 |
| *Methods based on the* **decoupled** *paradigm:* | | | | | | | | | | | | |
| MMRL (Guo & Gu, 2025) | 81.30 | 74.83 | 77.93 | 98.97 | 76.97 | 86.59 | 90.57 | 91.53 | 91.05 | 46.13 | 37.47 | 41.35 |
| w/o decoupling strategy | 81.30 | 70.00 | 75.23 | 98.97 | 73.03 | 84.04 | 90.57 | 89.90 | 90.23 | 46.13 | 35.10 | 39.87 |
| MMRL++ (Guo & Gu, 2026) | 81.23 | 75.23 | 78.11 | 98.50 | 77.47 | 86.73 | 90.50 | 91.70 | 91.10 | 46.47 | 38.50 | 42.11 |
| w/o decoupling strategy | 81.23 | 72.43 | 76.58 | 98.50 | 73.80 | 84.38 | 90.50 | 91.57 | 91.03 | 46.47 | 38.63 | 42.19 |
| *Methods based on the* **coupled prompt field**: | | | | | | | | | | | | |
| **AlignedNorm** | 81.70 | 73.83 | 77.57 | 98.40 | 76.03 | 85.78 | 90.57 | 91.60 | 91.08 | 46.20 | 38.60 | 42.06 |
| Δ | **+0.47** | **+1.40** | **+0.99** | -0.10 | **+2.23** | **+1.40** | **+0.07** | **+0.03** | **+0.05** | -0.27 | -0.03 | -0.13 |

| Method | SUN397 | | | DTD | | | EuroSAT | | | UCF101 | | |
|---|---|---|---|---|---|---|---|---|---|---|---|---|
| | Base | New | HM | Base | New | HM | Base | New | HM | Base | New | HM |
| CLIP (Radford et al., 2021) | 69.36 | 75.35 | 72.23 | 53.24 | 59.90 | 56.37 | 56.48 | 64.05 | 60.03 | 70.53 | 77.50 | 73.85 |
| *Methods based on the* **end-to-end** *paradigm:* | | | | | | | | | | | | |
| MaPLe (Khattak et al., 2023a) | 80.80 | 78.33 | 79.55 | 79.87 | 57.60 | 66.93 | 91.70 | 75.10 | 82.57 | 83.53 | 77.23 | 80.26 |
| PromptSRC (Khattak et al., 2023b) | 82.50 | 78.87 | 80.64 | 83.27 | 61.50 | 70.75 | 92.80 | 72.20 | 81.21 | 86.87 | 78.83 | 82.65 |
| HicroPL (Zheng et al., 2025) | 83.25 | 78.99 | 81.06 | 83.60 | 65.30 | 73.33 | 94.04 | 72.28 | 81.74 | 87.47 | 81.02 | 84.12 |
| *Methods based on the* **decoupled** *paradigm:* | | | | | | | | | | | | |
| MMRL (Guo & Gu, 2025) | 83.07 | 79.23 | 81.10 | 85.87 | 64.10 | 73.40 | 96.10 | 72.33 | 82.54 | 88.30 | 79.63 | 83.74 |
| w/o decoupling strategy | 83.07 | 76.63 | 79.72 | 85.87 | 59.77 | 70.48 | 96.10 | 59.20 | 73.27 | 88.30 | 76.30 | 81.86 |
| MMRL++ (Guo & Gu, 2026) | 82.93 | 79.57 | 81.22 | 85.07 | 65.83 | 74.22 | 95.73 | 84.17 | 89.58 | 87.37 | 80.43 | 83.76 |
| w/o decoupling strategy | 82.93 | 78.43 | 80.62 | 85.07 | 61.70 | 71.52 | 95.73 | 81.63 | 88.12 | 87.37 | 80.40 | 83.74 |
| *Methods based on the* **coupled prompt field**: | | | | | | | | | | | | |
| **AlignedNorm** | 82.93 | 79.13 | 80.99 | 84.63 | 65.23 | 73.67 | 96.10 | 86.63 | 91.12 | 87.43 | 81.00 | 84.09 |
| Δ | 0.00 | **+0.70** | **+0.37** | -0.44 | **+3.53** | **+2.15** | **+0.37** | **+5.00** | **+3.00** | **+0.06** | **+0.60** | **+0.35** |

egy, we select the two most competitive models (Guo & Gu, 2025; 2026). We further remove decoupled inference for these methods and evaluate them using the same base inference rule for a fair comparison. This leads to a clear drop in generalization performance. AlignedNorm boosts new-class accuracy without harming base-class performance, without decoupled inference. As shown in Fig. 4, Aligned-Norm yields more separable new-class embeddings.

AlignedNorm shows a slight drop on the new classes of FGVCAircraft. One possible reason is that its data distribution differs from CLIP's pretraining data, leading to a mismatch with CLIP's embedding space. Since our method tends to preserve CLIP's original space structure, the model may have less room to adapt in this case, which could limit learning and affect generalization.

Beyond standard accuracy metrics, we further report *uni-formity* and *tolerance* (Wang & Isola, 2020; Wang & Liu, 2021) to compare the **representation geometry** learned by MMRL++ and AlignedNorm. These metrics offer a complementary view of how norm alignment affects the learned embedding space, making the empirical behavior of AlignedNorm more interpretable and better aligned with our coupled prompt field analysis.

**Uniformity.** Let $\mathcal{D} = \{(f_i, y_i)\}_{i=1}^N$ denote the set of $\ell_2$-normalized embeddings for new classes. We measure uniformity via the Gaussian potential over pairwise distances:

$$\mathcal{L}_{\text{unif}} = \log \mathbb{E}_{i \neq j} \left[ \exp \left( -t \|f_i - f_j\|^2 \right) \right], \quad (13)$$

where we follow the default CLIP setting and set $t = 0.01$. A smaller $\mathcal{L}_{\text{unif}}$ indicates that embeddings are more evenly dispersed on the hypersphere, which typically corresponds to better separability among new classes.

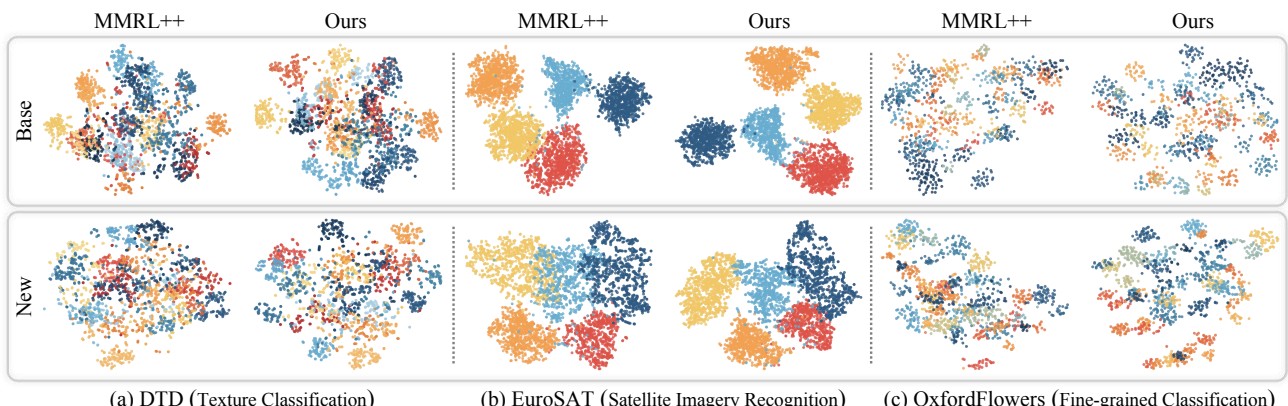

(a) DTD (Texture Classification)  (b) EuroSAT (Satellite Imagery Recognition)  (c) OxfordFlowers (Fine-grained Classification)

*Figure 4.* **t-SNE visualizations.** We visualize the embeddings of the decoupled method MMRL++ and AlignedNorm across three diverse image recognition datasets. AlignedNorm exhibits clearer separability among new classes.

*Table 2.* **Cross-dataset benchmark evaluation**. Notably, AlignedNorm achieves competitive performance without any test-time cues, demonstrating its robust generalization capabilities across diverse domains.

| | Source | Target | | | | | | | | | | |
|---|---|---|---|---|---|---|---|---|---|---|---|---|
| | ImageNet | Caltech101 | OxfordPets | StanfordCars | Flowers102 | Food101 | Aircraft | SUN397 | DTD | EuroSAT | UCF101 | Average |
| PromptSRC | 71.27 | 93.60 | 90.25 | 65.70 | 70.25 | 86.15 | 23.90 | 67.10 | **46.87** | 45.50 | 68.75 | 65.81 |
| MMRL | 71.93 | 94.40 | 91.20 | 65.67 | 72.90 | 86.23 | 26.43 | 67.40 | 46.50 | 52.27 | 69.03 | 67.20 |
| w/o decoupling | **71.93** | 94.17 | 89.53 | 63.40 | 71.77 | 84.87 | 24.80 | 65.80 | 46.40 | 53.10 | 68.40 | 66.22 |
| MMRL++ | 71.83 | 94.57 | 91.30 | 66.53 | 73.30 | 86.63 | 26.23 | 67.90 | 46.47 | 53.50 | 69.60 | 67.60 |
| w/o decoupling | 71.83 | 94.40 | 91.17 | 65.87 | 72.27 | 86.07 | 26.27 | 67.50 | 46.50 | **55.50** | 69.10 | 67.47 |
| **AlignedNorm** | 71.83 | **94.50** | **91.57** | **66.50** | **73.00** | **86.67** | **26.30** | **67.97** | 46.63 | 53.47 | **69.50** | **67.61** |

*Table 3.* **Few-shot learning.** AlignedNorm demonstrates highly competitive performance particularly in low-shot regimes. Detailed results can be found in Appendix 11.

| Method | 1 shot | 2 shots | 4 shots | 8 shots | 16 shots |
|---|---|---|---|---|---|
| Linear probe CLIP | 45.83 | 57.98 | 68.01 | 74.47 | 78.79 |
| PromptSRC | 72.27 | 75.24 | 78.20 | 80.55 | 82.76 |
| MMRL | 72.62 | **75.74** | 79.10 | **81.40** | **84.28** |
| MMRL++ | 72.57 | 75.35 | 78.95 | 81.30 | 84.07 |
| **AlignedNorm** | **72.93** | 75.57 | **79.11** | 81.35 | 84.08 |

**Tolerance.** Tolerance characterizes the local cohesion of embeddings within the same new class. With normalized embeddings, we compute it as the expected cosine similarity over same-class pairs:

$$\text{Tol} = \mathbb{E}_{i \neq j}\left[f_i^\top f_j \cdot \mathbb{I}(y_i = y_j)\right], \qquad (14)$$

where $\mathbb{I}(\cdot)$ is the indicator function. A higher Tol suggests stronger intra-class compactness, while overly optimizing uniformity may reduce tolerance by pushing semantically similar instances apart.

As shown in Fig. 5, we visualize the uniformity and toler-

ance of MMRL++ and AlignedNorm. On the new classes, AlignedNorm achieves uniformity closer to the original CLIP, suggesting that the coupled prompt field reshapes the embedding space in a globally coherent manner. Moreover, AlignedNorm attains tolerance comparable to or even higher than CLIP, whereas MMRL++ shows substantially lower tolerance, indicating that base-task knowledge is better transferred and generalizes to new tasks.

### 5.3. Cross Dataset Experiment

As shown in Table 2, AlignedNorm significantly improves cross-dataset generalization without relying on any decoupled strategy. In particular, AlignedNorm achieves consistent gains on 10 out of 11 datasets, demonstrating stronger robustness to distribution shifts across datasets.

### 5.4. Few-shot Experiment

In the few-shot setting, MMRL++ uses a single unified inference strategy across all shot numbers. As shown in Table 3, introducing norm alignment in AlignedNorm does not hurt performance; instead, the gains are more pronounced in lower-shot regimes. This pattern suggests that norm

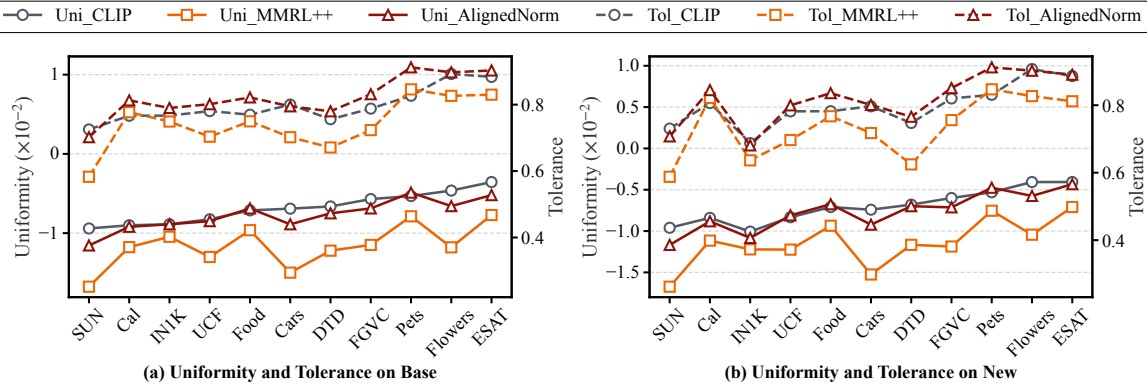

*Figure 5.* **Visualization of uniformity and tolerance**. While MMRL++ consistently lags behind CLIP in both metrics, Aligned-Norm achieves uniformity comparable to CLIP and further improves tolerance. These results highlight the superior potential and effectiveness of AlignedNorm in balancing these two key properties across diverse datasets, architectures, and evaluation settings.

*Table 4.* **Domain generalization.** All methods are trained on ImageNet and evaluated on 4 datasets with domain shifts.

| | Source | Target | | | | |
|---|---|---|---|---|---|---|
| | ImageNet | -V2 | -S | -A | -R | Avg. |
| CLIP | 66.73 | 60.83 | 46.15 | 47.77 | 73.96 | 57.17 |
| MMRL | **71.93** | 64.57 | 49.13 | 50.70 | 77.20 | 60.40 |
| w/o decoupling | **71.93** | **65.37** | 49.10 | 49.57 | 77.00 | 60.26 |
| MMRL++ | 71.83 | 64.37 | 49.23 | **51.10** | 77.63 | 60.58 |
| w/o decoupling | 71.83 | 65.27 | **49.53** | 50.80 | **78.10** | 60.93 |
| **AlignedNorm** | 71.83 | 65.13 | **49.53** | **51.10** | 78.07 | **60.96** |

*Table 5.* **Computational overhead.** Average training and inference time on ImageNet with a batch size of 32.

| Method | Training (s) | Testing (s) | Params (M) | HM |
|---|---|---|---|---|
| PromptSRC | **0.350** | 0.124 | 0.046 | 79.78 |
| HiCroPL | 0.362 | 0.163 | 0.246 | 80.60 |
| MMRL++ | 0.357 | **0.087** | **0.045** | 80.71 |
| **AlignedNorm** | 0.357 | **0.087** | **0.045** | **81.45** |

alignment mainly improves optimization stability when supervision is scarce, where the model is more vulnerable to overfitting and representation drift. Overall, these results underscore the role of norm alignment in preserving transferable knowledge.

### 5.5. Domain Generalization Experiment

We further evaluate domain generalization with domain shifts. Following prior practice, we train the model on ImageNet and directly evaluate it on the target datasets without additional adaptation. As summarized in Table 4, AlignedNorm remains competitive on the others, overall and consistently, indicating robust generalization under domain shifts and is particularly effective when transferring from

ImageNet to distribution-mismatched target datasets.

## 6. Discussion & Future Prospects

**Paradigm gap.** Prior studies have motivated decoupled prompt learning over end-to-end adaptation from channel-distribution (Zhang et al., 2024; 2026) and optimization (Li et al., 2025a) perspectives, showing the benefit of separating base-specific adaptation from pretrained knowledge preservation. However, decoupled methods require task identity at test time, leading to an inference gap. Inspired by the representation-geometry view based on uniformity and tolerance, as shown in Fig. 5, AlignedNorm bridges this inference gap through the coupled prompt field while still adhering to the core decoupled philosophy. Instead of modeling an ideal final representation that is difficult to define across base and new tasks, CPF models the more transferable prompt-induced change.

**Computational overhead.** AlignedNorm only adds a norm-alignment loss during training, whose computation is lightweight. As shown in Table 5, this introduces negligible overhead over MMRL++ in training time and does not increase the number of learnable parameters. Since the alignment loss is not used at inference time, Aligned-Norm has exactly the same testing cost as MMRL++. Thus, AlignedNorm retains the efficiency advantage of MMRL++ over other prompt-learning methods while improving its task-agnostic coupling behavior.

**Norm alignment ablation.** Table 6 presents the ablation results of our proposed norm alignment strategies. The results show that applying alignment after the projection stage consistently brings stable and reliable performance improvements across different classes. Furthermore, introducing additional alignment within each layer leads to further performance gains. However, using either one alone reduces base-class performance, especially layer-wise alignment by

*Table 6.* **Effect of different components.** Results are averaged over 11 datasets. HM refers to harmonic mean.

| Method | Base | New | HM |
|---|---|---|---|
| 1: MMRL++ | 85.43 | 76.48 | 80.71 |
| 2: $+ \mathcal{L}_{\text{token}}$ | 85.16 | 76.37 | 80.52 |
| 3: $+ \mathcal{L}_{\text{proj}}$ | 85.42 | 76.96 | 80.97 |
| 4: $+ \mathcal{L}_{\text{token}} + \mathcal{L}_{\text{proj}}$ | **85.46** | **77.79** | **81.45** |

*Table 7.* **Ablations on loss functions.** As illustrated, $L1$ loss provides better performance. HM refers to harmonic mean.

| Method | Base | New | HM |
|---|---|---|---|
| 1: MMRL++ | 85.43 | 76.48 | 80.71 |
| 2: + Hinge loss | **85.44** | 76.19 | 80.55 |
| 3: + smooth $L1$ loss | 85.32 | 76.83 | 80.85 |
| 4: + Ring loss | 85.29 | 76.94 | 80.90 |
| 5: + $L1$ loss | 85.42 | **76.96** | **80.97** |

itself, showing that the two levels need to work together in a complementary manner for the best results.

**Loss function ablation.** Table 7 compares different losses for norm alignment. Hinge loss aims to adjust the relative norm relationship between prompt tokens and class tokens, but it does not improve HM, suggesting that merely enforcing an inequality-style constraint is insufficient. Smooth-$L_1$ and Ring loss (Zheng et al., 2018) provide stronger norm-shrinkage or norm-matching effects, but neither matches the performance of $L_1$ alignment. This indicates that overly aggressive norm convergence may create numerical shortcuts, where the model focuses on satisfying the norm constraint rather than learning semantic features. It is also worth noting that Ring loss typically aligns feature norms to a fixed target, whereas our alignment dynamically uses the class token norm as a sample-dependent anchor. This dynamic anchoring better matches the coupled prompt field objective.

**Alignment weight selection.** Table 8 reports the alignment weights $(\beta, \gamma)$ used on the 11 datasets. The weights $\beta$ and $\gamma$ control the strengths of post-projection and layer-wise norm alignment, respectively. In practice, we follow two empirical principles. First, their values should keep the alignment loss on a comparable scale to the cross-entropy loss; overly large weights may cause the model to over-optimize norm matching and suppress semantic learning. Second, we set $\gamma \leq \beta$ so that the projected representation norm is stabilized before stronger constraints are imposed on intermediate prompt tokens. This hierarchy provides a top-down optimization signal for layer-wise alignment and empirically helps mitigate entanglement collapse.

**Limitations and future work.** Our coupled prompt field adopts a single-step optimization scheme, which is efficient and stable but may limit further gains in challenging set-

*Table 8.* **Alignment weight selection.** The values of $\beta$ and $\gamma$ used for each dataset in base-to-new generalization.

| Dataset | $\beta$ | $\gamma$ | Dataset | $\beta$ | $\gamma$ |
|---|---|---|---|---|---|
| ImageNet | 0.005 | 0.005 | UCF | 0.150 | 0.150 |
| EuroSAT | 0.100 | 0.050 | DTD | 0.200 | 0.150 |
| Caltech | 0.050 | 0.050 | SUN | 0.100 | 0.010 |
| Aircraft | 0.100 | 0.010 | Cars | 0.150 | 0.150 |
| Flowers | 0.150 | 0.100 | Pets | 0.100 | 0.010 |
| Food | 0.150 | 0.001 | | | |

tings. Future work will explore multi-step optimization and more adaptive norm-control strategies to improve robustness, adaptability, and training efficiency. This work focuses on norm dynamics in contrastive vision–language models such as CLIP, where image and text representations are aligned in a normalized embedding space. Whether similar norm-related effects arise in generative multimodal architectures remains an open question.

# 7. Conclusion

In this paper, we propose a Coupled Prompt Field paradigm for Base-to-New generalization and identify embedding norms as a key factor in stabilizing this field. Based on this insight, AlignedNorm enforces post-projection and layer-wise norm alignment to improve prompt learning. More broadly, our results highlight the value of meso-level representation properties, such as norm dynamics, rather than relying only on high-level regularization objectives or architecture-specific modifications, for efficient and robust model adaptation without catastrophic forgetting.

## Acknowledgement

This work was supported in part by the National Natural Science Foundation of China (No. 62476143), the Fundamental Research Funds for the Central Universities (Nankai University, No.63263250). We are grateful to Xian-Yu Zou, Long Chen, Ze-Wen Du, Bo-Wen Nie (NKU), Ge-Peng Ji (ANU), Yu-Kun Zhang (BIT), and Guan-Xiang Shen (RUC) for their insightful discussions.

# Impact Statement

This work aims to advance efficient and generalizable prompt learning for vision-language models. Our analysis highlights that embedding norms are not merely numerical artifacts but can affect representation geometry. This observation may be relevant to future studies of robustness, privacy leakage, or adversarial misuse, although this paper does not develop or evaluate such attacks. We encourage responsible deployment with careful data auditing, robustness evaluation, and privacy-aware safeguards.

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

# AlignedNorm: Prompting Vision-Language Models via Coupled Prompt Field
## Supplementary Material

## A. Additional Related Works

**Vision-Language Models** (VLMs), such as CLIP (Radford et al., 2021) and ALIGN (Jia et al., 2021), learn a shared embedding space for images and texts via a dual-tower architecture. Pre-trained on massive image–text pairs, they enable strong open-vocabulary transfer to a wide range of downstream tasks (Zhuge et al., 2021; Rao et al., 2022; Ji et al., 2023; Gao et al., 2024b;a; Zheng et al., 2024a; Jiang et al., 2024; Li et al., 2024a;b; Chen et al., 2025; Luo et al., 2025; Yan et al., 2025; Zhao et al., 2026; Wu et al., 2025b; Ji et al., 2025; 2026; Wang et al., 2025b; Huang et al., 2025; Dai et al., 2026; Chen et al., 2026; Wang et al., 2026). In practice, downstream inference and evaluation often operate in the cosine-similarity regime on $\ell_2$-normalized embeddings, emphasizing angular alignment on a modality-aligned hypersphere (Wang & Isola, 2020; Wang & Liu, 2021). By contrast, the geometry of the *unnormalized* embedding space–where phenomena such as modality gaps (Liang et al., 2022) and the cone effect (Liang et al., 2022; Schrodi et al., 2025) arise-has been less explored in downstream adaptation and analysis. Recent studies further suggest that this geometry, including embedding norms, is coupled with semantic properties (Desai et al., 2023; Chou & Alam, 2024; Levi & Gilboa, 2025), motivating a closer look at norm-related structure.

**Prompt Learning**, introduced from natural language processing (Jiang et al., 2020; Shin et al., 2020), aims to adapt vision-language models like CLIP to downstream tasks by adding continuous and learnable tokens to the sequence (Yin et al., 2025). CoOp (Zhou et al., 2022a;b) first adopted language-induced prompting and optimized textual context using task-specific data. However, restricting adaptation to the language branch of VLMs sharply degrades zero-shot performance on unseen categories. Subsequent efforts separately employ *(i) vision-language-hierarchical-prompting* (Zang et al., 2022; Khattak et al., 2023a; Zheng et al., 2025; Liu et al., 2025a), *(ii) network regularization* driven by intrinsic reference (Yao et al., 2023; Khattak et al., 2023b; Tian et al., 2024; Roy & Etemad, 2024; Xie et al., 2025), extrinsic reference (Hu et al., 2023; Zheng et al., 2024b; Ding et al., 2025; Li et al., 2025b), distribution diversity (Lu et al., 2022; Chen et al., 2023; Wu et al., 2025a; Cheng & Han, 2025), gradient alignment (Zhu et al., 2023; Li et al., 2023; Park et al., 2024; Liu et al., 2025b), base–new decoupling (Zhang et al., 2024; Li et al., 2025a), or *(iii) distillation* (Li et al., 2024c; Khattak et al., 2025) to safeguard the VLMs' generalization. Given the superior expressivity of deep over shallow prompting (Petrov et al., 2024), advanced methods (Guo & Gu, 2025; 2026) have adopted representation learning to deeper layers.

**Embedding Norm.** An embedding can be decomposed into its direction (semantic angle) and its norm (radial magnitude). While direction is widely prioritized, the embedding norm is often treated as a secondary or unimportant factor. Early unsupervised learning methods (Ranzato, 2009; Hariharan & Girshick, 2017; Zheng et al., 2018) impose norm constraints to avoid trivial collapse, whereas modern contrastive VLMs (Radford et al., 2021; Jia et al., 2021; Zhai et al., 2023; Chou & Alam, 2024) utilize cosine similarity by projecting embeddings onto a unit hypersphere, thereby bypassing the norm information. However, recent studies (Levi & Gilboa, 2025; Draganov et al., 2025; Zhang et al., 2025) show that embedding norms correlate with semantic characteristics.

**High-norm Tokens.** High-norm tokens can arise as an unintended training artifact in Transformers (Xiao et al., 2024; Darcet et al., 2024). Under causal masking, they may attract excessive attention and form "attention sinks" (Xiao et al., 2024), which can be alleviated at inference time by deactivating the relevant neurons (Yona et al., 2025). Without causal masking, they often emerge in low-entropy regions and aggregate global context (Darcet et al., 2024). Prior work mitigates these effects via register tokens (Darcet et al., 2024), light-weight fine-tuning (Wang et al., 2024), or test-time fixes (Nakamura et al., 2024; Jiang et al., 2025; Zhao et al., 2025).

## B. Derivation for Mathematical Analysis

### B.1. Non-uniform field updates

**Derivation of Eq.** (7). Assume the loss depends on the prompt branch only through the $\ell_2$-normalized embedding $f(z) = \frac{z}{\|z\|} \in \mathbb{R}^d$ (with $\|z\| > 0$). Denote the gradient w.r.t. the normalized embedding by $g \triangleq \frac{\partial \mathcal{L}}{\partial f} \in \mathbb{R}^d$. We derive the Jacobian of the normalization map.

Let $s = \|z\| = (z^\top z)^{1/2}$. Using differentials,

$$ds = d\|z\| = d(z^\top z)^{1/2} = \frac{1}{2}(z^\top z)^{-1/2} d(z^\top z) = \frac{1}{\|z\|} z^\top dz.$$

Noting $f = z/\|z\|$, one has $z^\top dz = \|z\| f^\top dz$, hence

$$ds = f^\top dz.$$

Now compute the differential of $f(z) = z/s$:

$$df = d\left(\frac{z}{s}\right) = \frac{1}{s} dz - \frac{z}{s^2} ds = \frac{1}{\|z\|} dz - \frac{z}{\|z\|^2}(f^\top dz).$$

Substituting $z = \|z\|f$ yields

$$df = \frac{1}{\|z\|}\left(I - ff^\top\right) dz.$$

Therefore, the Jacobian of the normalization map is

$$\frac{\partial f}{\partial z} = \frac{1}{\|z\|}\left(I - ff^\top\right),$$

where $P \triangleq I - ff^\top$ is an orthogonal projection matrix onto the tangent space of the unit sphere at $f$.

By the chain rule,

$$\frac{\partial \mathcal{L}}{\partial z} = \left(\frac{\partial f}{\partial z}\right)^\top \frac{\partial \mathcal{L}}{\partial f} = \frac{1}{\|z\|}\left(I - ff^\top\right) g,$$

since $I - ff^\top$ is symmetric. This gives Eq. (7). Moreover, because $P$ is a projection with operator norm $\|P\|_2 \leq 1$,

$$\left\|\frac{\partial \mathcal{L}}{\partial z}\right\| \leq \frac{1}{\|z\|}\|g\|,$$

showing that the effective gradient magnitude is inversely scaled by $\|z\|$.

**Derivation of Eq. (8).** Assume the stochastic gradient admits a decomposition

$$g = \bar{g} + \xi, \qquad \mathbb{E}[\xi] = 0,$$

where the expectation is taken over the stochasticity of the minibatch/optimizer (conditioning on the input $x$ and current parameters, so that $z$ and hence $f$ are fixed). Using Eq. (7) and writing $P = I - ff^\top$,

$$\nabla_z \mathcal{L} = \frac{1}{\|z\|} P(\bar{g} + \xi) = \underbrace{\frac{1}{\|z\|} P\bar{g}}_{\mathbb{E}[\nabla_z \mathcal{L}]} + \frac{1}{\|z\|} P\xi,$$

where $\mathbb{E}[\nabla_z \mathcal{L}] = \frac{1}{\|z\|} P\bar{g}$ follows from $\mathbb{E}[\xi] = 0$. Thus,

$$\nabla_z \mathcal{L} - \mathbb{E}[\nabla_z \mathcal{L}] = \frac{1}{\|z\|} P\xi.$$

Taking squared norms and using that $P$ is an orthogonal projector ($\|Pv\| \leq \|v\|$ for all $v$),

$$\left\|\nabla_z \mathcal{L} - \mathbb{E}[\nabla_z \mathcal{L}]\right\|^2 = \frac{1}{\|z\|^2}\|P\xi\|^2 \leq \frac{1}{\|z\|^2}\|\xi\|^2.$$

Finally, taking expectations yields

$$\mathbb{E}\left[\left\|\nabla_z \mathcal{L} - \mathbb{E}[\nabla_z \mathcal{L}]\right\|^2\right] \leq \mathbb{E}\left[\frac{\|\xi\|^2}{\|z\|^2}\right],$$

which is Eq. (8). In particular, when $\|z_p(x)\|$ is small, the factor $1/\|z_p(x)\|$ increases the effective update scale and amplifies the impact of stochastic gradient noise on the prompt branch.

## B.2. Field stability to perturbations.

**Derivation of Eq.** (9). Consider the prompt-branch pre-normalization feature under a stochastic perturbation

$$z_p = z_p^\star + \varepsilon, \qquad \mathbb{E}[\varepsilon \mid x] = 0,$$

where $z_p^\star$ denotes the (noise-free) feature and $\varepsilon$ models estimation noise due to finite data and stochastic optimization. Let the normalized prompt embedding be $f_p(z) = \frac{z}{\|z\|}$.

**Step 1: First-order approximation of $\ell_2$ normalization.** The Jacobian of the normalization map $f_p(z) = z/\|z\|$ at $z = z_p^\star$ is

$$J(z_p^\star) \triangleq \left.\frac{\partial f_p}{\partial z}\right|_{z=z_p^\star} = \frac{1}{\|z_p^\star\|}\left(I - f_p^\star f_p^{\star\top}\right),$$

where $f_p^\star \triangleq f_p(z_p^\star) = z_p^\star/\|z_p^\star\|$. Therefore, a first-order Taylor expansion gives

$$f_p - f_p^\star \approx J(z_p^\star)\varepsilon = \frac{1}{\|z_p^\star\|}\left(I - f_p^\star f_p^{\star\top}\right)\varepsilon.$$

**Step 2: Bounding the perturbation energy in the normalized space.** Let $P \triangleq I - f_p^\star f_p^{\star\top}$. Note that $P$ is an orthogonal projector onto the tangent space of the unit sphere at $f_p^\star$, hence it is symmetric and idempotent ($P^2 = P$), and satisfies $\|Pv\| \leq \|v\|$ for all $v$. Using the first-order approximation,

$$\|f_p - f_p^\star\|^2 \approx \left\|\frac{1}{\|z_p^\star\|}P\varepsilon\right\|^2 = \frac{1}{\|z_p^\star\|^2}\|P\varepsilon\|^2 \leq \frac{1}{\|z_p^\star\|^2}\|\varepsilon\|^2.$$

Taking conditional expectation yields

$$\mathbb{E}\left[\|f_p - f_p^\star\|^2 \mid x\right] \lesssim \frac{\mathbb{E}[\|\varepsilon\|^2 \mid x]}{\|z_p^\star(x)\|^2}.$$

**Step 3: Propagating to the coupled prompt field.** Recall that the coupled prompt field is defined as $\mathbf{u}_f(x) = \alpha(f_p(x) - f_c(x))$. Let its noise-free counterpart be $\mathbf{u}_f^\star(x) = \alpha(f_p^\star(x) - f_c(x))$, where $f_c(x)$ is treated as stable under anchoring (or, more generally, its perturbation is negligible compared with that of $f_p$). Then

$$\mathbf{u}_f(x) - \mathbf{u}_f^\star(x) = \alpha\big(f_p(x) - f_p^\star(x)\big),$$

and therefore

$$\|\mathbf{u}_f(x) - \mathbf{u}_f^\star(x)\|^2 = \alpha^2\|f_p(x) - f_p^\star(x)\|^2.$$

Taking conditional expectation and applying the bound from Step 2 gives

$$\mathbb{E}\left[\|\mathbf{u}_f(x) - \mathbf{u}_f^\star(x)\|^2 \mid x\right] \approx \alpha^2 \mathbb{E}\left[\|f_p(x) - f_p^\star(x)\|^2 \mid x\right]$$
$$\leq \alpha^2 \frac{\mathbb{E}[\|\varepsilon\|^2 \mid x]}{\|z_p^\star(x)\|^2}, \tag{15}$$

which matches Eq. (9).

**Remark.** The bound highlights a $1/\|z_p^\star(x)\|^2$ sensitivity: when the prompt-branch pre-normalization norm is small, the same level of perturbation energy $\mathbb{E}[\|\varepsilon\|^2 \mid x]$ leads to a larger deviation in the coupled field. Since anchoring keeps $f_c$ relatively stable, controlling the scale of $z_p$ improves the perturbation robustness of the coupled prompt field.

## B.3. High-dimensional misalignment

Let $u, v$ be independent random unit vectors uniformly sampled from the unit sphere in $\mathbb{R}^D$, and let $\theta = \arccos(u^\top v) \in [0, \pi]$ be their angle. By rotational symmetry, we may fix $u = e_1$, so $u^\top v = v_1$ (the first coordinate of $v$). Hence,

$$\mathbb{E}[u^\top v] = 0, \qquad \mathbb{E}\left[(u^\top v)^2\right] = \mathbb{E}[v_1^2] = \frac{1}{D}. \tag{16}$$

Therefore $\mathbb{E}[|u^\top v|] \leq \sqrt{\mathbb{E}[(u^\top v)^2]} = D^{-1/2}$, i.e., the dot product is typically close to 0. Since $\theta = \arccos(u^\top v)$, this implies that random directions are nearly orthogonal: $\theta$ concentrates around $\pi/2$. In particular, the distribution of $\theta$ is symmetric around $\pi/2$, hence

$$\mathbb{E}[\theta] = \frac{\pi}{2}. \tag{17}$$

Moreover, for any $\delta \in (0, \pi/2)$,

$$\mathbb{P}\big(|\theta - \tfrac{\pi}{2}| \geq \delta\big) = \mathbb{P}\big(|u^\top v| \geq \sin\delta\big) \leq 2\exp\Big(-\frac{(D-1)\sin^2\delta}{2}\Big), \tag{18}$$

showing that nontrivial alignment ($\theta$ far from $\pi/2$) becomes exponentially unlikely as $D$ grows.

### B.4. Entanglement Collapse

Global information exchange is important in various scenarios (Fan et al., 2020; 2021; Sun et al., 2024; Du et al., 2025a;b; Wang et al., 2025a; Gao et al., 2025; Hu et al., 2026). We provide a simple analysis showing that controlling token norms can prevent (and often reverse) prompt-patch entanglement collapse by keeping attention gating in a non-saturated regime and helping to preserve the information exchange.

**Setup.** Consider one self-attention head at some layer. For a fixed query token (a patch or class token), let its query vector be $q \in \mathbb{R}^D$ and let $\{k_i\}_{i=1}^N$ be the key vectors of all tokens in the same layer. Among them, $k_p$ denotes a prompt token key and the rest correspond to non-prompt tokens. Define attention weights

$$a_i = \frac{\exp(\ell_i)}{\sum_{j=1}^N \exp(\ell_j)}, \qquad \ell_i = \frac{1}{s} q^\top k_i, \tag{19}$$

where $s > 0$ is the usual scaling factor (e.g., $s = \sqrt{D}$). The attention output is $o = \sum_{i=1}^N a_i v_i$ with value vectors $\{v_i\}$.

**Lemma 1.** Let $\ell_{\max} = \max_j \ell_j$. Then for any token $i$,

$$a_i \leq \exp(\ell_i - \ell_{\max}), \qquad a_i \geq \frac{1}{N} \exp(\ell_i - \ell_{\max}). \tag{20}$$

*Proof.* The upper bound follows from $\sum_j \exp(\ell_j) \geq \exp(\ell_{\max})$. The lower bound follows from $\sum_j \exp(\ell_j) \leq N \exp(\ell_{\max})$. $\square$

**Lemma 2.** Let $\mathcal{L}$ be any loss depending on the attention output $o$. Then the gradient w.r.t. the prompt key $k_p$ satisfies

$$\left\| \frac{\partial \mathcal{L}}{\partial k_p} \right\| \leq \frac{1}{s} a_p \|q\| \left\| \frac{\partial \mathcal{L}}{\partial o} \right\| (\|v_p\| + \|o\|). \tag{21}$$

*Proof.* A standard differentiation of $o = \sum_i a_i v_i$ with $a = \mathrm{softmax}(\ell)$ yields

$$\frac{\partial \mathcal{L}}{\partial \ell_p} = \Big(\frac{\partial \mathcal{L}}{\partial o}\Big)^\top \frac{\partial o}{\partial \ell_p} = \Big(\frac{\partial \mathcal{L}}{\partial o}\Big)^\top \Big(a_p(v_p - o)\Big).$$

Using $\ell_p = \frac{1}{s} q^\top k_p$, one has $\frac{\partial \ell_p}{\partial k_p} = \frac{1}{s} q$ and thus

$$\frac{\partial \mathcal{L}}{\partial k_p} = \frac{\partial \mathcal{L}}{\partial \ell_p} \cdot \frac{\partial \ell_p}{\partial k_p} = \frac{1}{s} a_p \Big(\Big(\frac{\partial \mathcal{L}}{\partial o}\Big)^\top (v_p - o)\Big) q.$$

Taking norms and applying Cauchy–Schwarz gives (21). $\square$

**Implication: why isolation becomes self-reinforcing.** By Eq. (21), once $a_p$ becomes very small, the prompt key receives a vanishing gradient and is unlikely to rotate/align with the rest of tokens. Therefore, low attention on prompts is a *self-reinforcing* state: weak attention $\Rightarrow$ weak gradient $\Rightarrow$ prompts remain weakly coupled, which is precisely an entanglement collapse.

*Table 9.* Summary of the 15 datasets.

| Dataset | Classes | Train | Val | Test | Description | Prompt |
|---|---|---|---|---|---|---|
| ImageNet | 1000 | 1.28M | ~ | 50000 | Large-scale benchmark for object classification | "a photo of a [CLASS]." |
| Caltech101 | 100 | 4128 | 1649 | 2465 | Classic benchmark for object category recognition | "a photo of a [CLASS]." |
| OxfordPets | 37 | 2944 | 736 | 3669 | Fine-grained identification of cat and dog breeds | "a photo of a [CLASS], a type of pet." |
| StanfordCars | 196 | 6509 | 1635 | 8041 | Fine-grained recognition of car makes, models, and years | "a photo of a [CLASS]." |
| Flowers102 | 102 | 4093 | 1633 | 2463 | Fine-grained classification of flower species | "a photo of a [CLASS], a type of flower." |
| Food101 | 101 | 50500 | 20200 | 30300 | Fine-grained recognition of diverse food dishes | "a photo of [CLASS], a type of food." |
| FGVCAircraft | 100 | 3334 | 3333 | 3333 | Fine-grained classification of aircraft variants | "a photo of a [CLASS], a type of aircraft." |
| SUN397 | 397 | 15880 | 3970 | 19850 | Large-scale scene and environment understanding | "a photo of a [CLASS]." |
| DTD | 47 | 2820 | 1128 | 1692 | Classification of textures and visual patterns | "[CLASS] texture." |
| EuroSAT | 10 | 13500 | 5400 | 8100 | Land use & cover classification with satellite images | "a centered satellite photo of [CLASS]." |
| UCF101 | 101 | 7639 | 1898 | 3783 | Action recognition from video frames in realistic scenes | "a photo of a person doing [CLASS]." |
| ImageNetV2 | 1,000 | ~ | ~ | 10,000 | Test set for evaluating generalization on ImageNet distribution | "a photo of a [CLASS]." |
| ImageNet-Sketch | 1,000 | ~ | ~ | 50,889 | Object recognition under sketch-style domain shift | "a photo of a [CLASS]." |
| ImageNet-A | 200 | ~ | ~ | 7,500 | Robust object recognition on naturally adversarial ImageNet examples | "a photo of a [CLASS]." |
| ImageNet-R | 200 | ~ | ~ | 30,000 | Diverse artistic renditions of ImageNet object classes | "a photo of a [CLASS]." |

**Proposition.**  Write the prompt logit as $\ell_p = \frac{1}{s}\|q\|\,\|k_p\|\cos\theta_p$ and similarly $\ell_j = \frac{1}{s}\|q\|\,\|k_j\|\cos\theta_j$. Suppose (i) angles are not yet well learned in early training so that some $\cos\theta_p$ can be negative, and (ii) prompt norms can drift such that $\|k_p\|$ becomes much larger than the typical non-prompt norm. Then the logit gap $\Delta \triangleq \ell_{\max} - \ell_p$ can grow proportionally to $\|k_p\|$, and Eq. 20 implies

$$a_p \;\le\; \exp(-\Delta), \tag{22}$$

i.e., the prompt attention can be exponentially suppressed. Combining with Eq. (21), the prompt gradient is exponentially damped, making collapse hard to recover from.

In contrast, enforcing token-norm alignment (keeping $\|k_p\|$ comparable to $\|k_j\|$ across tokens/layers) controls the scale of logits and thus upper-bounds the possible gap $\Delta$. Consequently, $a_p$ is prevented from becoming exponentially small, preserving gradient flow to prompt tokens and maintaining prompt–patch entanglement.

> *Takeaway: Token norms act as a gating scale in dot-product attention. Norm alignment keeps attention logits in a learnable range, so prompts remain visible to other tokens and continue receiving meaningful gradients, which explains why directly aligning token norms can reverse entanglement collapse in practice.*

## C. Additional Implementation Details

Details of the 15 datasets are shown in Table 9. For a fair comparison with the baseline methods, we follow the hyperparameter settings of MMRL++ unless otherwise specified. Specifically, we set $\alpha$ to $0.7$ and activate deep prompts from layer $J = 6$. The visual anchoring weights used for each dataset are kept the same as those in MMRL++. We also follow the training schedule of MMRL++ for all datasets, except for SUN397, where we train for three additional epochs. All experiments are conducted with CLIP ViT-B/16 as the backbone. We use AdamW as the optimizer with a learning rate of $1 \times 10^{-3}$, a cosine learning-rate scheduler, and one warm-up epoch with a constant warm-up learning rate of $1 \times 10^{-5}$. The number of learnable prompt tokens is set to $5$.

## D. Additional Experimental Results

All base-to-new ablation experiments are presented in Table 10 and detailed results on all 11 datasets for few-shot learning are provided in Table 11. Fig. 6 and Fig. 7 show the attention maps of MMRL++ and AlignedNorm at different layers.

*Table 10.* **Base to new generalization ablation experiments.** The accuracy (%) on base and new classes for each dataset.

| Method | Average | | | ImageNet | | | Caltech101 | | | OxfordPets | | |
|---|---|---|---|---|---|---|---|---|---|---|---|---|
| | Base | New | HM | Base | New | HM | Base | New | HM | Base | New | HM |
| baseline | 85.43 | 76.48 | 80.71 | 77.60 | 71.30 | 74.32 | 98.90 | 94.57 | 96.69 | 95.43 | 96.87 | 96.14 |
| + $\mathcal{L}_{\text{token}}$ | 85.16 | 76.37 | 80.52 | 77.43 | 71.27 | 74.22 | 98.83 | **94.87** | **96.81** | 95.03 | 96.83 | 95.92 |
| + $\mathcal{L}_{\text{proj}}$(Hingeloss) | 85.44 | 76.19 | 80.55 | **77.67** | 71.23 | 74.31 | 98.90 | 94.57 | 96.69 | 95.63 | 97.07 | 96.34 |
| + $\mathcal{L}_{\text{proj}}$(smooth$L1$) | 85.32 | 76.83 | 80.85 | 77.47 | 71.33 | 74.27 | 98.73 | 94.63 | 96.64 | 95.17 | 96.83 | 95.99 |
| + $\mathcal{L}_{\text{proj}}$(Ring loss) | 85.29 | 76.94 | 80.90 | 77.60 | 71.30 | 74.32 | **98.93** | 94.70 | 96.77 | 95.63 | 97.37 | 95.99 |
| + $\mathcal{L}_{\text{proj}}(L1)$ | 85.42 | 76.96 | 80.97 | 77.57 | 71.27 | 74.29 | 98.87 | 94.40 | 96.58 | **95.67** | 97.23 | 96.44 |
| AlignedNorm | **85.46** | **77.79** | **81.45** | 77.60 | **71.47** | **74.41** | 98.90 | 94.77 | 96.79 | 95.63 | **97.43** | **96.52** |

| Method | StanfordCars | | | Flowers102 | | | Food101 | | | FGVCAircraft | | |
|---|---|---|---|---|---|---|---|---|---|---|---|---|
| | Base | New | HM | Base | New | HM | Base | New | HM | Base | New | HM |
| baseline | 81.23 | 72.43 | 76.58 | **98.50** | 73.80 | 84.38 | 90.50 | 91.57 | 91.03 | 46.47 | **38.63** | **42.19** |
| + $\mathcal{L}_{\text{token}}$ | 80.87 | 72.23 | 76.31 | 98.30 | 73.83 | 84.33 | 90.50 | 91.30 | 90.90 | 45.50 | 36.97 | 40.79 |
| + $\mathcal{L}_{\text{proj}}$(Hingeloss) | 80.97 | 71.17 | 75.75 | 98.20 | 73.47 | 84.05 | 90.53 | 91.30 | 90.91 | 45.87 | 37.20 | 41.08 |
| + $\mathcal{L}_{\text{proj}}$(smooth$L1$) | 80.67 | 73.23 | 76.77 | 98.27 | 75.03 | 85.09 | 90.57 | 91.37 | 90.97 | 46.43 | 37.83 | 41.69 |
| + $\mathcal{L}_{\text{proj}}$(Ring loss) | 80.53 | 73.53 | 76.87 | 98.07 | 75.90 | 85.57 | 90.67 | 91.50 | 91.08 | 46.10 | 37.37 | 41.28 |
| + $\mathcal{L}_{\text{proj}}(L1)$ | 80.70 | 73.53 | 76.95 | 98.27 | 75.90 | 85.65 | **90.73** | 91.60 | 91.16 | 46.10 | 38.47 | 41.94 |
| AlignedNorm | **81.70** | **73.83** | **77.57** | 98.40 | **76.03** | 85.78 | 90.57 | 91.60 | 91.08 | 46.20 | 38.60 | 42.06 |

| Method | SUN397 | | | DTD | | | EuroSAT | | | UCF101 | | |
|---|---|---|---|---|---|---|---|---|---|---|---|---|
| | Base | New | HM | Base | New | HM | Base | New | HM | Base | New | HM |
| baseline | 82.93 | 78.43 | 80.62 | 85.07 | 61.70 | 71.52 | 95.73 | 81.63 | 88.12 | 87.37 | 80.40 | 83.74 |
| + $\mathcal{L}_{\text{token}}$ | 83.03 | 78.43 | 80.66 | 84.80 | 63.53 | 72.64 | 95.63 | 80.80 | 87.59 | 86.80 | 80.00 | 83.26 |
| + $\mathcal{L}_{\text{proj}}$(Hingeloss) | 82.93 | 78.37 | 80.59 | **85.13** | 61.40 | 71.34 | 95.93 | 82.17 | 88.52 | **88.03** | 80.10 | 83.88 |
| + $\mathcal{L}_{\text{proj}}$(smooth$L1$) | 82.87 | 78.83 | 80.80 | 84.77 | 63.47 | 72.59 | **96.30** | 82.77 | 89.02 | 87.23 | 79.80 | 83.35 |
| + $\mathcal{L}_{\text{proj}}$(Ring loss) | 82.87 | 78.97 | 80.87 | 84.37 | 62.50 | 71.81 | 95.80 | 83.07 | 88.98 | 87.57 | 80.13 | 83.68 |
| + $\mathcal{L}_{\text{proj}}(L1)$ | **83.10** | 79.17 | 81.09 | 84.93 | 61.97 | 71.66 | 96.10 | 82.80 | 88.96 | 87.53 | 80.23 | 83.72 |
| AlignedNorm | 82.93 | 79.13 | 80.99 | 84.63 | **65.23** | **73.67** | 96.10 | **86.63** | **91.12** | 87.43 | **81.00** | **84.09** |

*Table 11.* Comparison of AlignedNorm with MMRL++ (Guo & Gu, 2026) on few-shot learning across 11 datasets.

| Dataset | Method | 1 shot | 2 shots | 4 shots | 8 shots | 16 shots |
|---|---|---|---|---|---|---|
| ImageNet | MMRL++ | 70.03 | 70.80 | 71.43 | 72.30 | **73.17** |
| | AlignedNorm | **70.07** | **70.87** | **71.47** | **72.33** | 73.07 |
| Caltech101 | MMRL++ | 94.13 | 94.87 | **95.90** | **96.13** | **96.83** |
| | AlignedNorm | **94.33** | **94.97** | 95.83 | 96.03 | **96.83** |
| OxfordPets | MMRL++ | **91.30** | **91.17** | 92.67 | 92.33 | **93.53** |
| | AlignedNorm | 91.17 | 90.93 | **92.87** | **92.53** | 93.50 |
| StanfordCars | MMRL++ | 68.30 | 72.57 | **77.97** | **82.43** | **86.20** |
| | AlignedNorm | **70.17** | **73.87** | 77.80 | 81.83 | 85.40 |
| OxfordFlowers | MMRL++ | 83.87 | **89.67** | **93.80** | 96.23 | **98.30** |
| | AlignedNorm | **84.53** | 89.63 | 93.73 | **96.33** | 97.87 |
| Food101 | MMRL++ | 82.93 | 84.03 | 84.63 | 85.43 | 86.13 |
| | AlignedNorm | **84.70** | **85.33** | **85.90** | **86.37** | **87.00** |
| FGVCAircraft | MMRL++ | 28.13 | 32.50 | 40.70 | **49.27** | **58.20** |
| | AlignedNorm | **28.30** | **33.83** | **40.80** | 48.80 | 58.03 |
| SUN397 | MMRL++ | 69.00 | 71.13 | 73.40 | 75.40 | 77.47 |
| | AlignedNorm | **69.50** | **71.43** | **73.97** | **75.87** | **77.63** |
| DTD | MMRL++ | **57.13** | 60.80 | **67.10** | 70.73 | 74.43 |
| | AlignedNorm | 56.80 | **60.90** | 66.80 | **71.00** | **74.80** |
| EuroSAT | MMRL++ | **78.00** | **82.40** | 88.33 | **89.30** | **93.33** |
| | AlignedNorm | 77.17 | 80.93 | **88.67** | 89.13 | 93.03 |
| UCF101 | MMRL++ | 75.50 | **78.90** | **82.50** | **84.73** | 87.23 |
| | AlignedNorm | **75.53** | 78.60 | 82.40 | 84.60 | **87.67** |

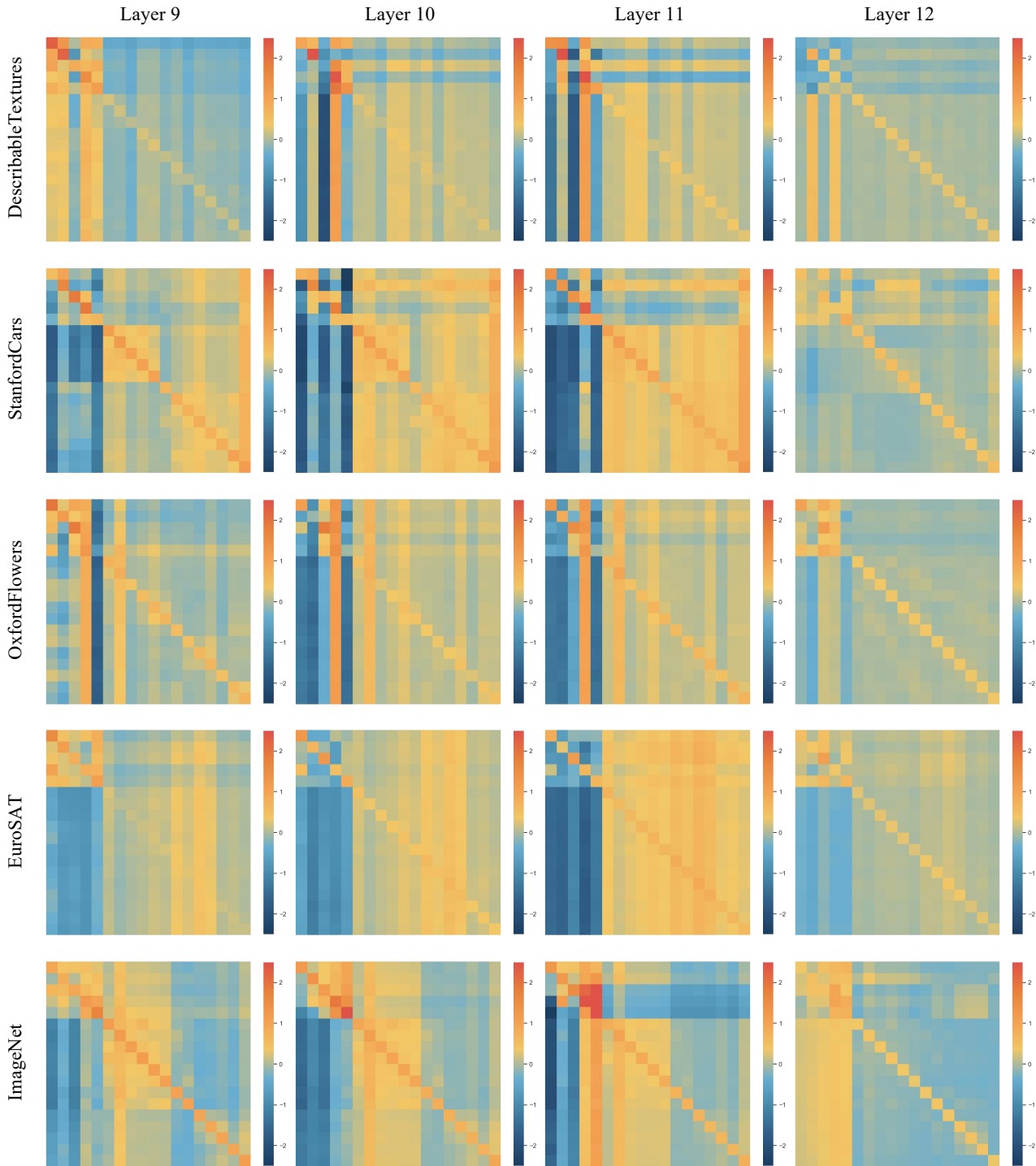

*Figure 6.* **The attention maps of the last four layers for MMRL++.**

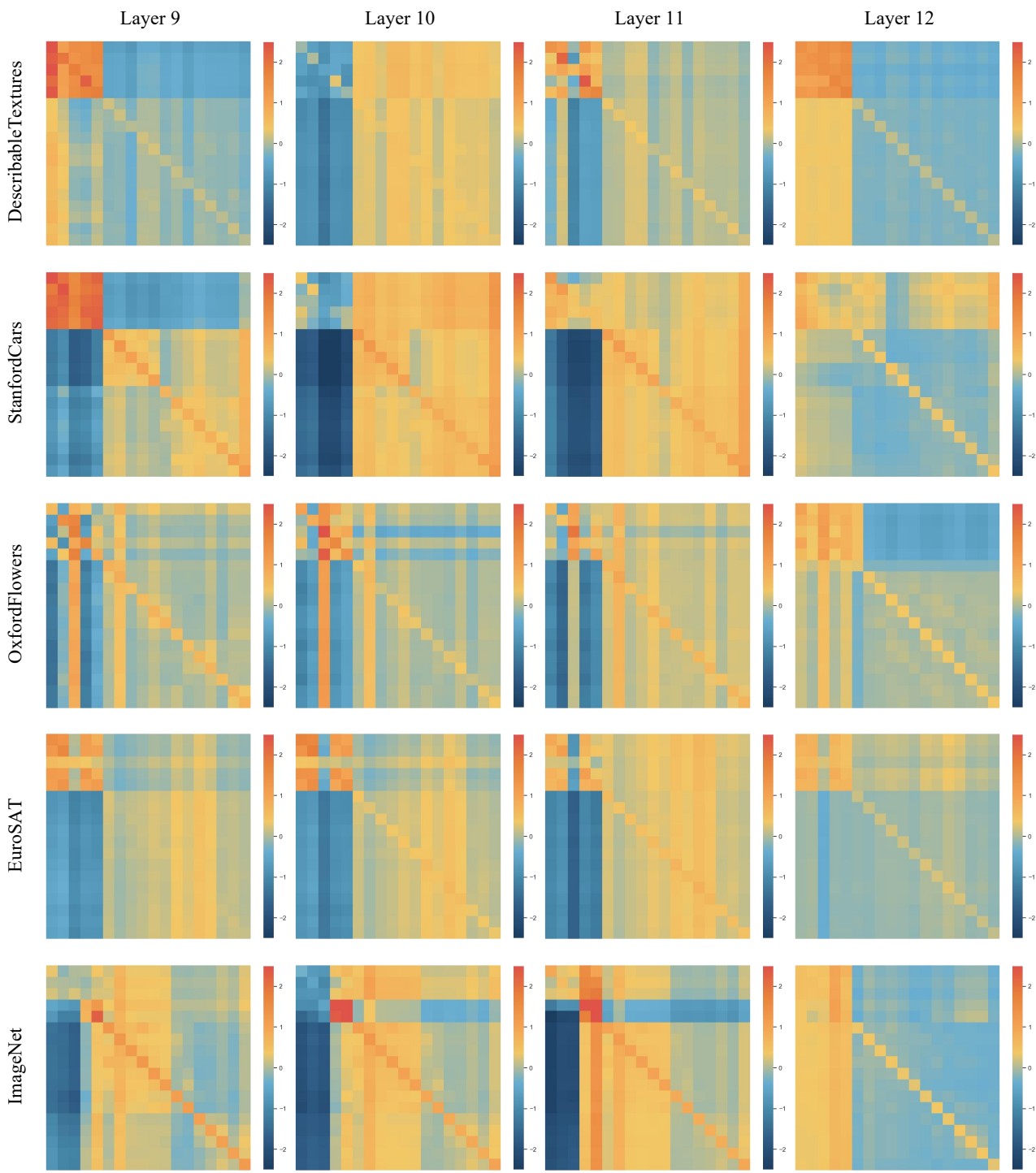

*Figure 7.* **The attention maps of the last four layers for AlignedNorm.**

