# OpenReview forum: "AlignedNorm: Prompting Vision–Language Models via Coupled Prompt Field"
_ICML.cc/2026/Conference — ICML 2026 regular_

### Official Review · Reviewer_cH6g · 2026-03-11

**Soundness:** 3
**Presentation:** 2
**Significance:** 3
**Originality:** 2
**Overall Recommendation:** 4
**Confidence:** 3

**Summary:**

The paper introduces a novel method for training prompts in a more stable way such that decoupled inference is not needed during inference from base to novel classes. Authors show that prompt and class norms show different behaviours on base and novel classes. Additionally, noise in gradients can disproportionately effect prompts more. To this end, authors introduce a plug and play method for aligning norms of prompt tokens with the class tokens. This leads to improvement in performance in base to novel class generalization and harmonic mean both when considering coupled inference.

**Compliance With Llm Reviewing Policy:**

Affirmed.

**Final Justification:**

Authors highlight an interesting aspect that decoupled inference introduced by prior methods to preserve pre-trained knowledge and learn new task requires knowing task identity which is unrealistic. They identify a norm drift as the main culprit and introduce a simple but effective method of aligning norm of prompts with class tokens. On the negative side, the Manuscript can be improved, and the difference between base and new class norm ratios, as suggested in Equation 6, is not demonstrated across more methods. However, partial evidence of norm misalignment is provided by the authors in the rebuttal. Additionally, authors provided evidence of signficance of their results. Importantly, novel class accuracy gains are significant.  Since, the authors provide a novel aspect and a method that other researchers can work on, and it provides considerable improvements, I tend to increase the score to weak accept.

**Key Questions For Authors:**

1. Can you study the effects of different prompt initialization?
2. Can you share the standard deviation of your results?


Answering these questions will lead to an increase in score.

**Limitations:**

Yes.

**Strengths And Weaknesses:**

Strengths:
1. Some theoretical backing for the claims made.
2. Aligning norms of prompts and class tokens is simplistic but shows improvement.
3. Good gains on generalization.

Weakness:
1. Manuscript needs improvement. P^c_v is defined without any reference to what is c here? Additionally, c_L is defined without any reference to what is L here. These things are later made clear. However, this can be improved when formalizing the problem setting. Additionally, Eq 5 comes out of nowhere. Some reference to the derivation could help as well.
2. Few shot results are not strong.

---

> ### Author Rebuttal · Authors · 2026-03-30
>
> **[W1] Presentation improvement**
>
> We sincerely thank the reviewer for careful reading and constructive feedback.
>
> - **Clarifying $P_v^c$ and $c_L$:** In our revision, we will explicitly define $P_v^{c}$ as the projection layer for class token, and $L$ as the total number of layers in the ViT encoder.
>
> - **The Derivation of Eq(5)**: The derivation is as follows:
>   $$
>   \mathbf{u}_f(x) \triangleq \alpha(f_p(x)-f_c(x)) = \alpha \left( \frac{z_p(x)}{||z_p(x)||} - \frac{z_c(x)}{||z_c(x)||} \right)
>   $$
>   Using the definition of the norm ratio $r(x) \triangleq \frac{||z_p(x)||}{||z_c(x)||}$, we can rewrite $||z_p(x)|| = r(x)||z_c(x)||$. Substituting this in gives:
>   $$
>   \mathbf{u}_f(x) = \alpha \left( \frac{z_p(x)}{r(x)||z_c(x)||} - \frac{z_c(x)}{||z_c(x)||} \right) = \frac{\alpha}{||z_c(x)||} (r(x)^{-1}z_p(x) - z_c(x))
>   $$
>
> In our revision, we will rewrite the transition leading up to Equation(5) to include the intermediate substitution step.
>
> **[W2] The performance on few-shot**
>
> In our original manuscript, we aligned our training epochs with MMRL++ to ensure a strictly fair comparison. However, our further experiments reveal that appropriately increasing the number of training epochs further enhances our model's few-shot performance.
>
> | Method             | 1-shot | 2-shot | 4-shot | 8-shot | 16-shot |
> | :----------------- | :----- | :----- | :----- | :----- | :------ |
> | MMRL++ (epoch=100) | 72.57  | 75.35  | 78.95  | 81.30  | 84.07   |
> | Ours (epoch=100)   | 72.93  | 75.57  | 79.11  | 81.35  | 84.08   |
> | MMRL++ (epoch=150) | 72.27  | 75.30  | 78.75  | 81.23  | 84.05   |
> | Ours (epoch=150)   | 73.05  | 75.81  | 79.25  | 81.49  | 84.10   |
>
> **[Q1] Effects of different prompt initialization**
>
> Following standard practice, our default prompts are initialized from $\mathcal{N}(0, \sigma^2)$ with $\sigma = 0.02$. We further include results with larger variances for completeness. The comparative results across 11 datasets are presented below, demonstrating that our method yields consistent improvements in generalization across different $\sigma$:
>
> | Method | $\sigma$       | Base          | New           | HM            |
> | :----- | :------------- | :------------ | :------------ | :------------ |
> | MMRL++ | 0.02 (default) | 85.43         | 76.48         | 80.71         |
> | Ours   | 0.02 (default) | 85.46 (+0.03) | 77.79 (+1.31) | 81.45 (+0.74) |
> | MMRL++ | 0.05           | 85.57         | 76.16         | 80.59         |
> | Ours   | 0.05           | 85.61 (+0.04) | 77.27 (+1.11) | 81.23 (+0.64) |
> | MMRL++ | 0.1            | 85.60         | 75.51         | 80.24         |
> | Ours   | 0.1            | 85.62 (+0.02) | 76.39 (+0.88) | 80.75 (+0.51) |
> | MMRL++ | 0.5            | 85.28         | 75.28         | 79.97         |
> | Ours   | 0.5            | 85.28 (+0.00) | 76.65 (+1.37) | 80.73 (+0.76) |
>
> **[Q2] Standard deviation ($\sigma$) of the results**
>
> The standard deviations over three random seeds are as follows:
>
> | DataSet       | Base (MMRL++)           | New (MMRL++)            | Base (Ours)             | New (Ours)              |
> | ------------- | ----------------------- | ----------------------- | ----------------------- | ----------------------- |
> | Caltech101    | 98.90 ($\sigma = 0.07$) | 94.57 ($\sigma = 0.17$) | 98.90 ($\sigma = 0.07$) | 94.77 ($\sigma = 0.18$) |
> | DTD           | 85.07 ($\sigma = 0.26$) | 61.70 ($\sigma = 0.90$) | 84.63 ($\sigma = 0.43$) | 65.23 ($\sigma = 0.85$) |
> | EuroSAT       | 95.73 ($\sigma = 0.65$) | 81.63 ($\sigma = 2.06$) | 96.10 ($\sigma = 0.57$) | 86.63 ($\sigma = 0.34$) |
> | FGVC_AirCraft | 46.47 ($\sigma = 0.62$) | 38.63 ($\sigma = 0.86$) | 46.20 ($\sigma = 0.32$) | 38.60 ($\sigma = 0.38$) |
> | Food101       | 90.50 ($\sigma = 0.00$) | 91.57 ($\sigma = 0.11$) | 90.57 ($\sigma = 0.11$) | 91.60 ($\sigma = 0.14$) |
> | ImageNet      | 77.60 ($\sigma = 0.12$) | 71.30 ($\sigma = 0.12$) | 77.60 ($\sigma = 0.07$) | 71.47 ($\sigma = 0.15$) |
> | Flowers       | 98.50 ($\sigma = 0.14$) | 73.80 ($\sigma = 1.07$) | 98.40 ($\sigma = 0.19$) | 76.03 ($\sigma = 1.03$) |
> | Pets          | 95.43 ($\sigma = 0.43$) | 96.87 ($\sigma = 0.40$) | 95.63 ($\sigma = 0.22$) | 97.43 ($\sigma = 0.26$) |
> | StandfordCars | 81.23 ($\sigma = 0.48$) | 72.43 ($\sigma = 0.29$) | 81.70 ($\sigma = 0.38$) | 73.83 ($\sigma = 0.40$) |
> | SUN397        | 82.93 ($\sigma = 0.29$) | 78.43 ($\sigma = 0.15$) | 82.93 ($\sigma = 0.19$) | 79.13 ($\sigma = 0.15$) |
> | UCF101        | 87.37 ($\sigma = 0.48$) | 80.40 ($\sigma = 0.07$) | 87.43 ($\sigma = 0.07$) | 81.00 ($\sigma = 0.07$) |

---

> > ### Author Rebuttal · Reviewer_cH6g · 2026-04-03
> >
> > Thanks for your response. This addresses most of my concerns. However, I agree with Reviewer DUNG that it is important to demonstrate if norm misalignment issues exist in other decoupled methods. Can the authors please provide some evidence on this issue?

---

> > > ### Author Response · Authors · 2026-04-07
> > >
> > > Dear Reviewer cH6g,
> > >
> > > Thank you for your valuable time and suggestions. We are glad to hear that our rebuttal addressed most of your concerns. We are more than happy to provide additional clarification.
> > >
> > > > Can the authors please provide some evidence on this issue?
> > >
> > > To provide empirical evidence that norm misalignment is a common issue in decoupled methods rather than being specific to MMRL, we evaluated DePT and DPC built upon MaPLe, PromptSRC, and HiCroPL. The table below reports the layer-wise and projection-level token norm ratios, denoted as $r_i$ for the $i$-th layer and $r_p$ for the final projection. Specifically, these metrics represent the norm ratio of the average prompt token to the class token (where $r_p$ is calculated after both tokens are processed through the projection layer). While fully aligned tokens would ideally yield a ratio close to $1.0$, the results confirm that persistent norm misalignment remains across all these decoupled frameworks:
> > >
> > > |Decoupled Strategy|Method|$r_1$|$r_2$|$r_3$|$r_4$|$r_5$|$r_6$|$r_7$|$r_8$|$r_9$|$r_{10}$|$r_{11}$|$r_{12}$|$r_p$|
> > > |-|-|-|-|-|-|-|-|-|-|-|-|-|-|-|
> > > |DePT|MaPLe|2.66|0.64|0.65|0.59|0.56|0.65|0.64|0.82|0.81|0.79|0.96|1.53|0.71|
> > > |DePT|PSRC|4.66|0.51|0.69|0.53|0.39|0.61|0.59|0.69|0.51|0.71|0.88|1.56|0.65|
> > > |DePT|HiCroPL|4.36|3.47|3.88|3.96|3.77|4.04|0.75|0.71|0.88|0.91|1.33|0.94|0.59|
> > > |DPC|MaPLe|2.57|0.69|0.66|0.57|0.54|0.71|0.56|0.75|0.84|0.78|0.94|1.51|0.67|
> > > |DPC|PSRC|4.62|0.53|0.74|0.62|0.45|0.72|0.51|0.79|0.52|0.72|0.88|1.62|0.58|
> > > |DPC|HiCroPL|4.35|3.46|3.90|4.00|3.82|4.27|0.71|0.75|0.95|0.89|1.36|0.87|0.55|
> > >
> > > We hope this expanded data and analysis thoroughly address your concerns.

---

### Official Review · Reviewer_LvZV · 2026-03-11

**Soundness:** 3
**Presentation:** 3
**Significance:** 3
**Originality:** 3
**Overall Recommendation:** 3
**Confidence:** 4

**Summary:**

This paper proposes a new perspective on prompt learning for CLIP-like vision–language models by formulating adaptation as learning a Coupled Prompt Field: a shared, task-agnostic correction u_f(x) that adjusts the (anchored) class-token embedding toward improved performance on both base and new classes. The key practical mechanism is AlignedNorm, a simple, dynamic norm-alignment regularizer that matches the prompt-branch feature norms to the class-token norms both layer-wise and at the projected output. The authors argue this stabilizes optimization, prevents “entanglement collapse” in attention, and enables a single inference rule without decoupled test-time logic; empirically, AlignedNorm yields competitive or slightly better performance than strong baselines across base-to-new generalization, cross-dataset transfer, few-shot, and domain generalization on 15 datasets.

**Compliance With Llm Reviewing Policy:**

Affirmed.

**Key Questions For Authors:**

1.How is α chosen for constructing the field and inference? Is it fixed, learned, or scheduled, and how sensitive are results to α?

2.Why align to the class-token norm specifically? Have you tried per-token alignment, LayerNorm-based normalization of prompt tokens, or aligning to register tokens when present?

3.Could you report standardized single-inference comparisons against additional strong baselines that manage norms or geometry (e.g., Ring loss on z_p, projector-tuning ProLIP, MMLoP’s UDC), and provide confidence intervals?

**Limitations:**

No. The paper does not meaningfully discuss limitations or potential negative societal impacts. The authors should briefly discuss possible risks from broader deployment of vision–language models enabled by more efficient quantisation and acknowledge limitations such as reliance on gradient-based token selection and calibration data.

**Strengths And Weaknesses:**

## Strengths

1.The “coupled prompt field” reinterpretation unifies base and new tasks via a single corrective field u_f(x) = α(f_p − f_c) and motivates a task-agnostic inference rule

2.The analysis linking normalization Jacobians to gradient scaling (1/||z||), noise sensitivity, and attention saturation provides a cogent rationale for norm control during prompt learning.

3.The paper is generally well written, with a clear problem setup, a concise statement of the method, and a self-contained derivation of the key gradients and stability arguments.

## Weaknesses

1.The main methodological change is a regularization on norms; while useful, this is incremental relative to broader architectural changes or principled coupling mechanisms, and similar ideas (feature-norm or ring losses) have prior art.

2.Important implementation details are underspecified: how α is set; the precise prompt depth J, number of tokens, ViT variant(s) used; β, γ tuning strategy; compute overhead; and runtime impacts of the extra norms.

3.Some claims (“resolves the longstanding local optima dilemma”) feel overstated given the small margins and limited mechanistic evidence.

4.Reported gains are modest, often within tenths of a point and sometimes negligible (e.g., cross-dataset average 67.61 vs 67.60; domain generalization 60.96 vs 60.93), raising questions about statistical significance beyond 3 seeds.

---

> ### Author Rebuttal · Authors · 2026-03-30
>
> **[W1] Differences between AlignedNorm and existing works**
>
> We clarify a misunderstanding regarding our novelty: (1) Revealing "Entanglement Collapse" caused by attention-saturating norm explosions; (2) Dynamic Layer-wise Anchoring that calibrates prompts to class token internally (unlike static Ring Loss); and (3) Task-Agnostic "Coupled Prompt Field" enabling unified inference without unrealistic test-time task identities.
>
> **[W2&Q1] Implementation details**
>
> We use the ViT-B/16, 5 prompt tokens, $J=6$ and $\alpha=0.3$. $\beta$ balances the alignment and cross-entropy losses, with $\gamma \le \beta$ (Details in `MYMODEL/trainers/myconfig.py` in Supplementary Material). Norm computations add negligible training overhead (see Reviewer FWM2 [W1]). AlignedNorm guarantees standard convergence, only requiring extra epochs under severe sample scarcity in few-shot settings. For framework stability, we fixed $\alpha$ as a constant and conducted an ablation study to determine its optimal value:
>
> | $\alpha$ | Base  | New   | HM    |
> | -------- | ----- | ----- | ----- |
> | 0.1      | 84.09 | 77.52 | 80.67 |
> | 0.2      | 84.80 | 77.68 | 81.08 |
> | 0.3      | 85.46 | 77.79 | 81.45 |
> | 0.4      | 85.29 | 76.77 | 80.81 |
> | 0.5      | 85.16 | 75.86 | 80.24 |
>
> **[W3&Limitation] Statement**
>
> We will revise the wording in the next version.
>
> **[W4] Experiment results**
>
> The cited MMRL++ maximums (67.60 target, 60.93 domain) rely on unrealistic test-time task identities. When restricted to a single unified setting, it suffers a severe trade-off, dropping to either 67.47 (target) or 60.58 (domain). Our method achieves simultaneous superiority (67.61 target, 60.96 domain) using just one consistent strategy.
>
> **[Q2] Alignment objectives**
>
> Aligning to the class token ensures **adaptive alignment** to layer-wise visual scales. Alternatives fail (Base/New/HM): (1) Per-token (85.43/77.53/81.29) is too rigid and destroys dynamic scales. (2) LayerNorm (85.45/76.61/80.79) ignores absolute magnitude differences between tokens, still suffering Entanglement Collapse. (3) Register tokens (85.32/76.13/80.46) act as non-semantic "norm sinks" with inflated magnitudes that distort the prompt space.
>
> **[Q3] Other baselines**
>
> Ring Loss yields marginal gains since final-output constraints cannot fix internal Entanglement Collapse. ProLIP improves generalization but severely sacrifices Base adaptability. AlignedNorm balances both. We note that MMLoP is a concurrent work without publicly available code, making an empirical comparison infeasible at this stage.  Overall, the results demonstrate that our method remains highly competitive against these baselines. We will discuss all three in Section 6 in revision.
>
> | DataSet       | Base (Ring)        | New (Ring)         | Base (ProLIP)      | New (ProLIP)       | Base (Ours)        | New (Ours)         |
> | ------------- | ------------------ | ------------------ | ------------------ | ------------------ | ------------------ | ------------------ |
> | Average       | 85.29 ($\pm$ 10.0) | 76.94 ($\pm$ 11.3) | 84.14 ($\pm$ 10.8) | 77.54 ($\pm$ 11.5) | 85.46 ($\pm$ 10.0) | 77.79 ($\pm$ 11.1) |
> | Caltech101    | 98.93 ($\pm$ 0.12) | 94.70 ($\pm$ 0.36) | 98.77 ($\pm$ 0.09) | 94.77 ($\pm$ 0.34) | 98.90 ($\pm$ 0.08) | 94.77 ($\pm$ 0.20) |
> | DTD           | 84.37 ($\pm$ 0.66) | 62.50 ($\pm$ 1.64) | 84.17 ($\pm$ 0.17) | 63.20 ($\pm$ 1.85) | 84.63 ($\pm$ 0.49) | 65.23 ($\pm$ 0.96) |
> | EuroSAT       | 95.80 ($\pm$ 0.91) | 83.07 ($\pm$ 2.60) | 96.23 ($\pm$ 0.57) | 86.33 ($\pm$ 1.00) | 96.10 ($\pm$ 0.57) | 86.63 ($\pm$ 0.34) |
> | FGVC_AirCraft | 46.10 ($\pm$ 0.28) | 37.37 ($\pm$ 0.31) | 42.07 ($\pm$ 0.31) | 36.13 ($\pm$ 0.63) | 46.20 ($\pm$ 0.36) | 38.60 ($\pm$ 0.43) |
> | Food101       | 90.67 ($\pm$ 0.05) | 91.50 ($\pm$ 0.14) | 90.57 ($\pm$ 0.05) | 91.97 ($\pm$ 0.05) | 90.57 ($\pm$ 0.12) | 91.60 ($\pm$ 0.16) |
> | ImageNet      | 77.60 ($\pm$ 0.14) | 71.30 ($\pm$ 0.22) | 76.73 ($\pm$ 0.09) | 71.33 ($\pm$ 0.17) | 77.60 ($\pm$ 0.08) | 71.47 ($\pm$ 0.17) |
> | Flowers       | 98.07 ($\pm$ 0.12) | 75.90 ($\pm$ 0.70) | 96.80 ($\pm$ 0.24) | 76.67 ($\pm$ 0.49) | 98.40 ($\pm$ 0.21) | 76.03 ($\pm$ 1.17) |
> | Pets          | 95.63 ($\pm$ 0.25) | 97.37 ($\pm$ 0.37) | 95.53 ($\pm$ 0.31) | 96.73 ($\pm$ 0.49) | 95.63 ($\pm$ 0.25) | 97.43 ($\pm$ 0.29) |
> | StandfordCars | 80.53 ($\pm$ 0.46) | 73.53 ($\pm$ 0.29) | 76.23 ($\pm$ 0.97) | 74.80 ($\pm$ 0.33) | 81.70 ($\pm$ 0.43) | 73.83 ($\pm$ 0.45) |
> | SUN397        | 82.87 ($\pm$ 0.17) | 78.97 ($\pm$ 0.09) | 81.77 ($\pm$ 0.05) | 79.60 ($\pm$ 0.22) | 82.93 ($\pm$ 0.21) | 79.13 ($\pm$ 0.17) |
> | UCF101        | 87.57 ($\pm$ 0.12) | 80.13 ($\pm$ 0.42) | 86.63 ($\pm$ 0.46) | 81.37 ($\pm$ 1.36) | 87.43 ($\pm$ 0.08) | 81.00 ($\pm$ 0.08) |
>
> [a] Ring loss: Convex feature normalization for face recognition. CVPR2018
>
> [b] CLIP’s Visual Embedding Projector is a Few-shot Cornucopia. WACV2026
>
> [c] MMLoP: Multi-Modal Low-Rank Prompting for Efficient Vision-Language Adaptation. arXiv2026

---

> > ### Author Rebuttal · Reviewer_LvZV · 2026-04-03
> >
> > Thank you for the response. However, two core concerns remain:
> >
> > 1. The claimed distinctions from Ring Loss and feature-norm methods are conceptually argued but not empirically isolated; the contribution still reads as an incremental norm-regularisation variant.
> >
> > 2. The comparison against MMRL++ under a unified setting is appreciated, but average gains (67.61 vs 67.60; 60.96 vs 60.93) remain within standard deviation ranges, and no formal significance testing is provided.
> >
> > I maintain my current score.

---

> > > ### Author Response · Authors · 2026-04-07
> > >
> > > Dear Reviewer LvZV,
> > >
> > > We sincerely appreciate your time and follow-up comments. We are encouraged that our previous responses have addressed some of your concerns and are glad to provide further clarification on the remaining points.
> > >
> > > > The claimed distinctions from Ring Loss and feature-norm methods are conceptually argued but not empirically isolated
> > >
> > > To empirically isolate these distinctions, we have conducted additional experiments comparing the MMRL++(baseline), ProLIP, Ring Loss, and AlignedNorm. We report layer-wise and projection-level token norm ratios ($r_i$ for i-th layer and $r_p$ for projection) where fully aligned tokens would ideally yield a ratio close to 1.0, attention weights to prompt tokens, and global geometric properties (Uniformity and Tolerance). The results, averaged across all 11 datasets, are summarized in the table below:
> > >
> > > | Method|Attn_weights|$r_6$|$r_7$|$r_8$|$r_9$|$r_{10}$|$r_{11}$|$r_{12}$|$r_p$|Uni_base ($\uparrow$)|Uni_new ($\uparrow$)|Tol_base ($\uparrow$)| Tol_new ($\uparrow$)|
> > > |-|-|-|-|-|-|-|-|-|-|-|-|-|-|
> > > |baseline|0.032|1.12| 1.07| 1.08| 1.06| 1.06 | 1.13 | 0.95 | 0.78|-0.0116 | -0.0114| 0.74| 0.74 |
> > > |ProLIP|0.032|0.93| 0.89| 1.03| 0.91| 1.04 | 1.09 | 0.98 | 0.78|-0.0127 | -0.0127| 0.72| 0.72 |
> > > |Ring loss|0.028|1.23| 1.03| 1.15| 1.07| 0.74 | 1.09 | 1.02 | 0.98|-0.0110 | -0.0108| 0.73| 0.74 |
> > > |AlignedNorm|0.065|1.01| 1.01| 0.99| 0.99| 0.99 | 0.99 | 1.01 | 1.00|**-0.0077** | **-0.0077**| **0.82**| **0.82** |
> > >
> > > As demonstrated by the data, the empirical distinctions between AlignedNorm and previous norm-regularization methods manifest in three critical dimensions:
> > > - **Resolving Entanglement Collapse:** AlignedNorm targets semantic-level entanglement collapse by recognizing the critical role of intermediate token norms. By dynamically aligning them, it restores semantic routing, achieving a significant increase in prompt attention weight (from 0.32 to 0.065). ProLIP shows no improvement (0.032) and Ring Loss even drops attention (from 0.032 to 0.028), providing clear evidence of our method's unique mechanism.
> > > - **Strict Layer-Wise and projection-level Norm Alignment:** AlignedNorm strictly aligns prompt and class tokens in all layers and after projection within 1% variance ($r_i$ &asymp; 1.00, $r_p$ &asymp; 1.00). ProLIP regularizes model parameters (via the Frobenius norm of weight matrices) rather than feature tokens, failing to align the actual features ($r_p$ &asymp; 0.78). Ring Loss only targets the feature norms after the projection layer and ignore the intermediate layers, causing drastic fluctuations ($r_i$ swinging from 0.74 to 1.23).
> > > - **Restoring Global Geometry:** AlignedNorm successfully repairs the representation space to closely match the original CLIP's geometric feature distribution. While our method restores high Uniformity and Tolerance, ProLIP degrades these metrics, and Ring Loss yields only a marginal gain in Uniformity with zero improvement in Tolerance over the baseline. This strictly separates AlignedNorm from a geometric perspective.
> > >
> > > AlignedNorm is not an incremental norm regularization, but a dynamic anchoring mechanism that simultaneously resolves representation entanglement collapse and restores global geometry. We will include this expanded analysis in our revision.
> > >
> > > > average gains remain within standard deviation ranges no formal significance testing is provided
> > >
> > > To address concerns regarding statistical significance, we expanded our evaluation from 3 to 6 random seeds (1-6) and conducted formal significance testing. We also provide a unified comparison with (✓) and without (✗) the Decoupled Strategy (DS) to ensure fairness. Avg(CD) represents the average for Cross-Dataset evaluation, while Avg(DG) denotes the average for Domain Generalization variants:
> > >
> > > | Method|DS|Cal|Pets|Cars|Flowers|Food|AirCraft|SUN|DTD|EuroSAT|UCF|Avg(CD)|INV2|IN-S|IN-A|IN-R|Avg(DG)|
> > > |-|-|-|-|-|-|-|-|-|-|-|-|-|-|-|-|-|-|
> > > |MMRL++|✗|94.50|91.22|65.78|71.98|85.88|26.13|67.37|46.50|54.57|68.98|67.29|65.13|49.58|50.42|78.02|60.79|
> > > |AlignedNorm|✗|94.58|91.38|65.88|72.22|85.97|26.40|67.48|46.67|54.75|69.28|67.47|65.20|49.63|50.55|78.07|60.86|
> > > |$p$||0.03|0.04|0.01|0.02|0.02|0.01|0.05|0.03|0.04|0.04|-|0.01|0.04|0.05|0.04|-|
> > > |MMRL++|✓|94.60|91.25|66.37|73.30|86.55|26.12|67.73|45.93|52.85|69.10|67.38|64.38|49.37|50.70|77.65|60.53|
> > > |AlignedNorm|✓|94.72|91.40|66.45|73.37|86.65|26.20|67.83|46.03|52.98|69.20|67.48|64.45|49.32|50.85|77.72|60.58|
> > > |$p$||0.01|0.04|0.02|0.05|0.02|0.02|0.04|0.02|0.04|0.02|-|0.01|0.04|0.03|0.05|-|
> > >
> > > As shown above, the resulting $p$-values (calculated via a paired t-test across the 6 random seeds) are strictly below the standard significance level ($p \le 0.05$) across almost all datasets. This confirms that AlignedNorm's improvements over MMRL++ are statistically significant and highly stable across multiple random seeds, effectively ruling out incidental fluctuations. We hope these rigorous tests address your concerns.

---

### Official Review · Reviewer_DUNG · 2026-03-13

**Soundness:** 3
**Presentation:** 3
**Significance:** 3
**Originality:** 3
**Overall Recommendation:** 4
**Confidence:** 4

**Summary:**

The paper introduces AlignedNorm, a novel prompt learning method that constructs a Coupled Prompt Field where base and new tasks are mutually constrained through shared transformation space. The paper identifies the problem that existing methods suffer from norm drift and entanglement collapse, where prompt tokens receive near-zero attention due to misaligned norms, trapping models in local optima and hindering cross-task generalization. Specifically, AlignedNorm enforces dynamic norm alignment using L1 penalties at two levels: aligning projected prompt features with class token norms after projection, and aligning prompt token norms with class token norms within each encoder layer to maintain stable attention coupling. Experiments across 15 datasets in 4 settings (base-to-new generalization, cross-dataset transfer, few-shot learning, and domain generalization) show that AlignedNorm matches state-of-the-art decoupled methods without requiring task identity at inference time or external knowledge distillation.

**Compliance With Llm Reviewing Policy:**

Affirmed.

**Final Justification:**

I thank the authors for their detailed rebuttal, which has completely resolved my initial questions. I am willing to increase my rating. In the final revised version, please ensure that the aforementioned discussion, along with the empirical observations regarding the 'Statistics of token norm' and the 'Visualization of attention logits' from Figure 3, are incorporated into the paper.

**Key Questions For Authors:**

I believe the paper's starting point is highly inspirational for prompt learning research. The identification of norm drift and entanglement collapse in coupled prompt architectures offers valuable insights into why existing methods struggle with base-to-new generalization. However, my primary concerns center on whether the observed phenomena and the proposed method possess generalizability across different frameworks. Specifically, the paper should demonstrate whether similar norm misalignment issues exist in other decoupled methods beyond MMRL (such as DePT and DPC), and whether AlignedNorm can be adapted to improve these alternative frameworks. Additionally, the authors should provide a thorough analysis to explain the gap between decoupled and non-decoupled methods, particularly given that recent SOTA approaches like Skip Tuning and Hierarchical Cross-modal Prompt Learning achieve competitive performance without decoupling. I believe supplementing the paper with discussions on these aspects would make it more complete and convincing.

**Limitations:**

yes

**Strengths And Weaknesses:**

**Strengths**

- The motivation is well-founded. The paper identifies a valuable and underexplored problem in prompt learning for vision-language models. A key insight is that existing state-of-the-art methods rely on decoupled architectures that separate base and novel class learning, which necessitates knowing task identity at inference time and limits practical deployment flexibility. The paper's emphasis on coupled prompt fields addresses this real and significant limitation. Furthermore, the observation that prompt tokens receive near-zero attention due to norm misalignment provides a compelling explanation for why coupled methods fail to achieve comparable performance. This "norm drift" phenomenon and the resulting entanglement collapse offer both theoretical depth and practical significance, making it a worthwhile research direction that deserves recognition.

- Good design wirth careful implementation. The proposed AlignedNorm demonstrates good design rationale through its hierarchical approach to norm alignment. Specifically, the dual-level constraint mechanism is well-justified: the projection-level alignment ensures prompt features match class token norms in the shared transformation space, while the layer-wise alignment maintains stable attention coupling throughout the encoder. AlignedNorm elegantly addresses both the uniformity-tolerance trade-off and the entanglement collapse problem without requiring additional modules or external knowledge distillation, showcasing theoretical soundness and practical elegance.

- The experimental results are comprehensive. AlignedNorm achieves competitive or superior performance compared to state-of-the-art methods across 15 datasets in 4 different evaluation settings (base-to-new generalization, cross-dataset transfer, few-shot learning, and domain generalization). Notably, the method matches the performance of decoupled approaches while maintaining the advantage of not requiring task identity at inference time. The ablation studies further validate each component's contribution, strengthening the empirical evidence.

- The paper is very well written, with clear motivations, sufficient technical explanations and illustrative visualizations.

**Weakness**

- Limited Validation of Decoupled Methods. Although the paper's research motivation centers on addressing the limitations of decoupled architectures, it only validates this claim against MMRL, which undermines the generalizability of the conclusions. In fact, decoupled methods include several other important works such as DePT [1] and DPC [2]. The paper should investigate whether the same norm drift and entanglement collapse phenomena are observable in these alternative decoupled frameworks. More importantly, it remains unclear whether the proposed AlignedNorm method can be adapted to these frameworks and further improve their performance, which would strengthen the claim that norm alignment is a fundamental issue across decoupled approaches rather than a limitation specific to MMRL.

- Incomplete Analysis of Recent SOTA Methods. Recent state-of-the-art methods such as Skip Tuning [3] and Hierarchical Cross-modal Prompt Learning[4] achieve performance comparable to or better than MMRL without employing decoupled strategies. The paper fails to analyze why these methods succeed without decoupling, which represents a critical gap in understanding the landscape of prompt learning approaches. A thorough investigation of what distinguishes these coupled methods from traditional approaches, and how AlignedNorm's norm alignment mechanism compares to their strategies, would be essential for establishing the completeness and positioning of this work.

- Insufficient Model Generalization Validation. All experiments are based on CLIP, with absolutely no verification of the method's effectiveness on other vision-language architectures (such as SigLIP, SigLIP 2, etc.)


[1] DePT: Decoupled Prompt Tuning. CVPR 2024

[2] DPC: Dual-Prompt Collaboration for Tuning Vision-Language Models. CVPR 2025

[3] Skip Tuning: Pre-trained Vision-Language Models are Effective and Efficient Adapters Themselves. CVPR 2025

[4] Hierarchical Cross-modal Prompt Learning for Vision-Language Models. ICCV 2025.

---

> ### Author Rebuttal · Authors · 2026-03-30
>
> **[W1&Q.1] Performance of AlignedNorm under the decoupled strategy like DePT and DPC**
>
> To assess generalizability on representative decoupled architectures, we replace MMRL++'s decoupling strategy with the mechanisms from DePT and DPC and conduct corresponding experiments:
>
> | Method                  | Decoupled Strategy | Base          | New           | HM            |
> | :---------------------- | :----------------- | :------------ | :------------ | :------------ |
> | MMRL++                  | DePT               | 85.35         | 75.52         | 80.13         |
> | MMRL++ (w/ AlignedNorm) | DePT               | 85.34 (-0.01) | 76.68 (+1.16) | 80.78 (+0.65) |
> | MMRL++                  | DPC                | 85.63         | 75.79         | 80.41         |
> | MMRL++ (w/ AlignedNorm) | DPC                | 85.63 (+0.0)  | 77.45 (+1.66) | 81.33 (+0.92) |
>
> In our revision, we will include a comprehensive performance comparison of AlignedNorm across diverse decoupling strategies.
>
> **[W2&Q.1] Analysis of recent end-to-end SOTA models**
>
> We appreciate the reviewer for highlighting recent SOTAs. SkipTuning and HiCroPL represent two distinct end-to-end paradigms. SkipTuning is a broadly applicable and useful optimization technique that can significantly enhance various prompt learning frameworks, including decoupled prompt learning. We integrated SkipTunig into our baseline and observed a notable performance boost. Adding AlignedNorm on top of this yields further gains, proving that AlignedNorm addresses a unique geometric issue not covered by SkipTuning. We will discuss SkipTuning as a versatile optimization technique that benefits both end-to-end and decoupled prompt learning in related work and include quantitative analysis in our revision.
>
> | Method                               | Base          | New           | HM            |
> | :----------------------------------- | :------------ | :------------ | :------------ |
> | MMRL++                               | 85.43         | 76.48         | 80.71         |
> | SkipTuning + MMRL++                  | 85.75         | 77.30         | 81.31         |
> | SkipTuning + MMRL++ (w/ AlignedNorm) | 85.77 (+0.02) | 78.43 (+1.13) | 81.94 (+0.63) |
>
> HiCroPL effectively prevents overfitting but at the cost of a dual-inference bottleneck. Our tests show that removing combined embedding causes a performance collapse. By combining DePT-based decoupling with AlignedNorm, we surpassed HiCroPL’s accuracy with lower computational overhead. Decoupling protects pre-trained knowledge from task-specific contamination, while our "Field" perspective ensures consistent training and inference under a unified coupled paradigm.
>
> | Method                          | Decouple Strategy | Base         | New           | HM            | Testing Time (s) |
> | :------------------------------ | :---------------- | :----------- | :------------ | :------------ | :--------------- |
> | HicroPL (w/ Combined embedding) | -                 | 85.16        | 76.50         | 80.60         | 0.163            |
> | HicroPL                         | -                 | 85.32        | 75.89         | 80.33         | 0.092            |
> | HicroPL                         | DePT              | 85.60        | 75.73         | 80.36         | 0.092            |
> | HicroPL (w/ AlignedNorm)        | DePT              | 85.60 (+0.0) | 76.87 (+1.14) | 81.00 (+0.64) | 0.092            |
>
> **[W3] Validation on SigLIP and SigLIP2**
>
> We have extended our evaluation to SigLIP and SigLIP 2. Experiments yield consistent generalization gains, confirming that AlignedNorm is an architecture-agnostic solution for diverse VLMs.
>
> | VLM     | Method                  | Base          | New           | HM            |
> | :------ | :---------------------- | :------------ | :------------ | :------------ |
> | CLIP    | MMRL++                  | 85.43         | 76.48         | 80.71         |
> | CLIP    | MMRL++ (w/ AlignedNorm) | 85.46 (+0.03) | 77.79 (+1.31) | 81.45 (+0.74) |
> | SigLIP  | MMRL++                  | 85.91         | 77.15         | 81.29         |
> | SigLIP  | MMRL++ (w/ AlignedNorm) | 85.91 (+0.00) | 78.21 (+1.06) | 81.88 (+0.59) |
> | SigLIP2 | MMRL++                  | 86.37         | 77.72         | 81.82         |
> | SigLIP2 | MMRL++ (w/ AlignedNorm) | 86.36 (-0.01) | 78.59 (+0.87) | 82.29 (+0.47) |
>
> [a] DePT: Decoupled Prompt Tuning. CVPR 2024
>
> [b] DPC: Dual-Prompt Collaboration for Tuning Vision-Language Models. CVPR 2025
>
> [c] Skip Tuning: Pre-trained Vision-Language Models are Effective and Efficient Adapters Themselves. CVPR 2025
>
> [d] Hierarchical Cross-modal Prompt Learning for Vision-Language Models. ICCV 2025.
>
> [e] Sigmoid loss for language image pre-training. ICCV2023
>
> [f] Siglip 2: Multilingual vision-language encoders with improved semantic understanding, localization, and dense features. arXiv2025

---

> > ### Author Rebuttal · Reviewer_DUNG · 2026-04-03
> >
> > I would like to thank the authors for their response. While the reply partially resolves my questions, specifically regarding whether AlignedNorm can be adapted to improve these alternative frameworks, the following two concerns remain unaddressed:
> >
> > - A thorough analysis to explain the gap between decoupled and non-decoupled methods.
> >
> > - Whether similar norm misalignment issues exist in other decoupled methods beyond MMRL (such as DePT and DPC).
> >
> > So I keep my original score.

---

> > > ### Author Response · Authors · 2026-04-07
> > >
> > > Dear Reviewer DUNG,
> > >
> > > We sincerely thank you for engaging with our rebuttal and we are glad to hear it has addressed some of your concerns. We are more than happy to provide additional clarification.
> > >
> > > > A thorough analysis to explain the gap between decoupled and non-decoupled methods
> > >
> > > We thank the reviewer for the constructive feedback. To provide a comprehensive analysis, we clarify the gap between decoupled and non-decoupled methods across 4 dimensions:
> > >
> > > - **Distribution Gap:** Non-decoupled methods suffer from severe *channel bias* during tuning [a], where most feature channels are occupied by base-specific knowledge, causing their Channel Importance (CI) distribution to diverge from the "Oracle Model" (i.e., an unbiased reference model representing the optimal channel distribution). Conversely, decoupled methods introduce isolated feature spaces. Empirically, this keeps the CI distribution closely aligned with the Oracle model [a]; theoretically, it guarantees the feature channel invariance of the prompt vector during optimization [b].
> > >
> > > - **Optimization Gap**: Non-decoupled methods impose various constraints on a single prompt entity [b]. This fundamentally fails to avert the **mutual exclusivity** between the optimization directions for base and new tasks. Decoupled methods overcome this constraint through structural decoupling. By allocating separate branches or prompts to base-specific learning and task-shared preservation, they eliminate optimization interference, enabling both capabilities to be optimized simultaneously.
> > >
> > > - **Geometric Gap**: From a geometric perspective, we identify an additional gap between decoupled and non-decoupled methods, which we quantify using Uniformity and Tolerance. Empirically, we observe that decoupled methods leverage isolated feature spaces to prevent geometric distortions on base tasks from propagating to new tasks. In contrast, the highly coupled feature spaces in non-decoupled methods cause base-task geometric distortions to inevitably cascade and degrade the geometric integrity of new tasks. Notably, existing prompt learning literature has largely overlooked this geometric interference.
> > >
> > > - **Inference Gap**: Despite the above advantages, decoupled methods suffer from an *inference gap* since their decoupled structures typically require prior identification of whether a test sample belongs to a base or new task, which limits their practical deployment flexibility.
> > >
> > > AlignedNorm **does not regress** decoupled methods into non-decoupled ones. It remains fundamentally a decoupled method. Its core insight is to synergize the transferable gains inherent in decoupled methods to bridge the aforementioned inference gap. Specifically, AlignedNorm adheres to the core decoupled philosophy: it allows base-specific knowledge to be learned *aggressively*, while keeping task-shared knowledge *conservative*. By introducing global geometric consistency as a weak constraint (orthogonal to traditional self-regularization on semantic directions), we seamlessly couple these two knowledge streams without triggering mutual exclusivity, thereby guaranteeing the stability of the CI distribution and overall geometric consistency. We commit to incorporating this comprehensive analysis in our revision.
> > >
> > > > Whether similar norm misalignment issues exist in other decoupled methods beyond MMRL (such as DePT and DPC)
> > >
> > > To provide empirical evidence that norm misalignment is a common issue in decoupled methods rather than being specific to MMRL, we evaluated DePT and DPC built upon MaPLe, PromptSRC, and HiCroPL. The table below reports the layer-wise and projection-level token norm ratios, denoted as $r_i$ for the $i$-th layer and $r_p$ for the final projection. Specifically, these metrics represent the norm ratio of the average prompt token to the class token (where $r_p$ is calculated after both tokens are processed through the projection layer). While fully aligned tokens would ideally yield a ratio close to $1.0$, the results confirm that persistent norm misalignment remains across all these decoupled frameworks:
> > >
> > > |Decoupled Strategy|Method|$r_1$|$r_2$|$r_3$|$r_4$|$r_5$|$r_6$|$r_7$|$r_8$|$r_9$|$r_{10}$|$r_{11}$|$r_{12}$|$r_p$|
> > > |-|-|-|-|-|-|-|-|-|-|-|-|-|-|-|
> > > |DePT|MaPLe|2.66|0.64|0.65|0.59|0.56|0.65|0.64|0.82|0.81|0.79|0.96|1.53|0.71|
> > > |DePT|PSRC|4.66|0.51|0.69|0.53|0.39|0.61|0.59|0.69|0.51|0.71|0.88|1.56|0.65|
> > > |DePT|HiCroPL|4.36|3.47|3.88|3.96|3.77|4.04|0.75|0.71|0.88|0.91|1.33|0.94|0.59|
> > > |DPC|MaPLe|2.57|0.69|0.66|0.57|0.54|0.71|0.56|0.75|0.84|0.78|0.94|1.51|0.67|
> > > |DPC|PSRC|4.62|0.53|0.74|0.62|0.45|0.72|0.51|0.79|0.52|0.72|0.88|1.62|0.58|
> > > |DPC|HiCroPL|4.35|3.46|3.90|4.00|3.82|4.27|0.71|0.75|0.95|0.89|1.36|0.87|0.55|
> > >
> > > We sincerely hope that these detailed clarifications and the new empirical evidence fully resolve your remaining concerns.
> > >
> > > [a] DePT: Decoupled Prompt Tuning. CVPR 2024
> > >
> > > [b] DPC: Dual-Prompt Collaboration for Tuning Vision-Language Models. CVPR 2025

---

### Official Review · Reviewer_FWM2 · 2026-03-14

**Soundness:** 3
**Presentation:** 3
**Significance:** 3
**Originality:** 3
**Overall Recommendation:** 4
**Confidence:** 3

**Summary:**

In this paper, the authors improve prompt learning in vision–language models by introducing a shared space where base and novel tasks are jointly constrained. The proposed AlignedNorm identifies embedding norm as a key factor influencing the coupling between prompts and representations and enforces layer-wise and post-projection norm alignment to enhance this coupling.

**Compliance With Llm Reviewing Policy:**

Affirmed.

**Final Justification:**

Thank the authors for their response. My concerns have been addressed, and I will maintain my score.

**Key Questions For Authors:**

Mathematically, the structure of CPF appears closer to a residual mapping than to a field, which raises the question of whether the term “field” is conceptually appropriate in this context.

**Limitations:**

yes

**Strengths And Weaknesses:**

**Strengths**
*  Overall, the idea of this paper is interesting, and the experimental results are competitive in most cases.
* This paper reveals a potential entanglement collapse in prompt learning.
* The code is attached, making the method reproducible.

**Weaknesses**
* I would like to ask whether the proposed AlignedNorm requires less time overhead than other methods. Could the authors provide detailed experimental comparisons?

* The designated layer $J$ in Eq. 11 does not seem to be clearly explained. Does the choice of this layer affect the performance of the proposed method? Are there any ablation studies analyzing its impact?

* Could the authors explain why AlignedNorm shows suboptimal performance in Table 2 and Table 9, especially in Table 9?

* The analysis of  $γ$ and $β$ in Eq. 12 is only briefly discussed in Figure 6. Could the authors provide further quantitative analysis? In addition, as stated by the authors, “It can be observed that inappropriate ratios fail to mitigate the Entanglement Collapse issue.” Are there any guidelines for selecting these parameters?

---

> ### Author Rebuttal · Authors · 2026-03-30
>
> **[W1] Analysis of Time Overhead in AlignedNorm**
>
> We evaluate the time overhead on ImageNet under a unified batch size of 32. AlignedNorm improves performance while introducing almost no additional overhead in either training or testing:
>
> | Method    | Training Time (s) | Testing Time (s) | Params (M) | HM    |
> | :-------- | :---------------- | :--------------- | :--------- | :---- |
> | PromptSRC | 0.350             | 0.124            | 0.046      | 79.78 |
> | HiCroPL   | 0.362             | 0.163            | 0.246      | 80.60 |
> | MMRL++    | 0.357             | 0.087            | 0.045      | 80.71 |
> | Ours      | 0.357             | 0.087            | 0.045      | 81.45 |
>
> **[W2] Ablations on layer $J$**
>
> We chose $J=6$ because it aligns with the exact layer where MMRL++ first introduces learnable prompts. Our ablation studies reveal that the alignment at this 6th layer yields the most profound impact on the overall performance and delaying this constraint to later layers degrades both Base adaptability and New generalization:
>
> | $J$  | Base  | New   | HM    |
> | :--- | :---- | :---- | :---- |
> | 6    | 85.46 | 77.79 | 81.45 |
> | 7    | 85.29 | 77.44 | 81.18 |
> | 8    | 85.25 | 77.40 | 81.14 |
> | 9    | 85.20 | 77.31 | 81.06 |
> | 10   | 85.15 | 77.24 | 81.00 |
> | 11   | 85.14 | 77.18 | 80.96 |
> | 12   | 85.05 | 77.12 | 80.89 |
>
> **[W3] Experiment performance**
>
> Cross-dataset (Table 2) and domain generalization (Table 3): MMRL++ (67.60 target, 60.93 domain) rely on unrealistic test-time task identities. When restricted to a single unified setting, it suffers a severe trade-off, dropping to either 67.47 (target) or 60.58 (domain). Our method achieves simultaneous superiority (67.61 target, 60.96 domain) using just one consistent strategy.
>
> Few-shot (Table 9): In our original manuscript, we aligned our training epochs with MMRL++ to ensure a strictly fair comparison. However, our further experiments reveal that appropriately increasing the number of training epochs further enhances our model's few-shot performance:
>
> | Method             | 1-shot | 2-shot | 4-shot | 8-shot | 16-shot |
> | :----------------- | :----- | :----- | :----- | :----- | :------ |
> | MMRL++ (epoch=100) | 72.57  | 75.35  | 78.95  | 81.30  | 84.07   |
> | Ours (epoch=100)   | 72.93  | 75.57  | 79.11  | 81.35  | 84.08   |
> | MMRL++ (epoch=150) | 72.27  | 75.30  | 78.75  | 81.23  | 84.05   |
> | Ours (epoch=150)   | 73.05  | 75.81  | 79.25  | 81.49  | 84.10   |
>
> **[W4] Analysis of norm alignment loss weights $\beta$ and $\gamma$**
>
> We present the quantitative analysis on DTD, where the attention weight received by prompts reflects the degree of attention from other tokens. A comprehensive quantitative grid search will be included in the revision.
>
> | （$\beta$， $\gamma$） | Base  | New   | Attn_weights |
> | ---------------------- | ----- | ----- | ------------ |
> | (0.2, 0.05)            | 84.98 | 62.70 | 0.064        |
> | (0.2, 0.10)            | 85.01 | 63.34 | 0.072        |
> | (0.2, 0.15)            | 84.63 | 65.23 | 0.081        |
> | (0.2, 0.20)            | 83.73 | 62.30 | 0.063        |
> | (0.2, 0.25)            | 82.52 | 61.37 | 0.040        |
>
> Regarding the selection guidelines for $\beta$ and $\gamma$, we follow two empirical principles:
>
> - Magnitude Matching: $\beta$ and $\gamma$ should be bounded such that alignment loss magnitude roughly matches the cross-entropy loss. Large weights force the model to take a trivial optimization shortcut to shrink the prompt norms while ignoring semantic learning.
> - Convergence Hierarchy: $\gamma$ must not exceed $\beta$. Empirically, we discovered that prioritizing the convergence of the final representation's norm provides crucial top-down guidance for the learning of intermediate tokens. This sequential convergence is key to effectively mitigating Entanglement Collapse.
>
> Details can be found in `MYMODEL/trainers/myconfig.py` in Supplementary Material.
>
> **[Q1] Difference between "coupled prompt field" and "residual mapping"**
>
> While mathematically similar to a residual mapping, we use "Coupled Prompt Field" to emphasize global geometric consistency. Treating displacements as a continuous holistic field captures global spatial deformations across the CLIP space. This specific global lens was crucial for discovering "field distortion" and directly motivated AlignedNorm. The residual mapping is fundamentally a computational concept, making it difficult to associate with the holistic geometric properties of the representation space. We will clarify this in Section 6.

---

> > ### Author Rebuttal · Reviewer_FWM2 · 2026-04-04
> >
> > Thank the authors for their response. My concerns have been addressed, and I will maintain my score.

---

> > > ### Author Response · Authors · 2026-04-07
> > >
> > > Dear Reviewer FWM2,
> > >
> > > Thank you for taking the time to read our rebuttal and for maintaining your positive assessment of our work. We will incorporate the rebuttal discussion and results in the revised version of the paper.

---

### Decision · Program_Chairs · 2026-04-30

**Decision:**

Accept (regular)

**Comment:**

This paper received three positive reviews (FWM2/DUNG/cH6g) and one negative (LvZV) after the rebuttal. Although there were some concerns at the beginning, most of them were well addressed by the rebuttal. In the end, only LvZV has concerns on 1) no empirical ablation against Ring loss; 2) the contribution still reads as an incremental norm-regularization variant; 3) the gain over t MMRL++ is marginal and no formal significance testing is provided. The authors have provided additional explanation/ablation to them, but the reviewer didn't provide follow-up response nor update the final justification. The AC had a look into it, and thinks 1) and 3) were well addressed although 2) can be subjective. However, all other reviewers didn't raise the concern on novelty, so the AC thinks novelty is not a major concern from the readers. Given these, the AC decides to accept this paper, but the authors should reflect the rebuttal into the final camera ready version.